# Cell-specific measurements show nitrogen fixation by particle-attached putative non-cyanobacterial diazotrophs in the North Pacific Subtropical Gyre

Katie J. Harding[1,2], Kendra A. Turk-Kubo [1], Esther Wing Kwan Mak[1], Peter K. Weber [2], Xavier Mayali [2] ✉ & Jonathan P. Zehr [1] ✉

Biological nitrogen fixation is a major important source of nitrogen for low-nutrient surface oceanic waters. Nitrogen-fixing (diazotrophic) cyanobacteria are believed to be the primary contributors to this process, but the contribution of non-cyanobacterial diazotrophic organisms in oxygenated surface water, while hypothesized to be important, has yet to be demonstrated. In this study, we used simultaneous $^{15}N$-dinitrogen and $^{13}C$-bicarbonate incubations combined with nanoscale secondary ion mass spectrometry analysis to screen tens of thousands of mostly particle-associated, cell-like regions of interest collected from the North Pacific Subtropical Gyre. These dual isotope incubations allow us to distinguish between non-cyanobacterial and cyanobacterial nitrogen-fixing microorganisms and to measure putative cell-specific nitrogen fixation rates. With this approach, we detect nitrogen fixation by putative non-cyanobacterial diazotrophs in the oxygenated surface ocean, which are associated with organic-rich particles (<210 μm size fraction) at two out of seven locations sampled. When present, up to 4.1% of the analyzed particles contain at least one active putative non-cyanobacterial diazotroph. The putative non-cyanobacterial diazotroph nitrogen fixation rates ($0.76 \pm 1.60$ fmol N cell$^{-1}$ d$^{-1}$) suggest that these organisms are capable of fixing dinitrogen in oxygenated surface water, at least when attached to particles, and may contribute to oceanic nitrogen fixation.

Primary productivity in the oceans is commonly limited by nutrient availability, and nitrogen (N) is the limiting nutrient in large regions of the surface oceans[1]. Biological nitrogen fixation is an energetically expensive process that converts atmospheric dinitrogen ($N_2$) to bioavailable forms of N (ammonia and amino acids) that support primary production[2] and 26–47% of particulate N export[3]. Quantifying biological $N_2$ fixation rates is critical for predicting carbon (C) and N fluxes, yet there is high variability among biogeochemical model estimates of

$N_2$ fixation, sometimes as much as several orders of magnitude[4], suggesting these processes are not well-constrained. This disconnect is likely due to many factors but two related, yet distinct factors will be mentioned here. First, only a few $N_2$-fixing microorganisms are included in these models, and there are likely many other organisms yet to be discovered that fix $N_2$ in the surface ocean[5]. Second, the physiology, genetic diversity, and in situ activity of the known diazotrophs are not well constrained, making them difficult to model at the ocean

[1]Department of Ocean Sciences, University of California, Santa Cruz, CA, USA. [2]Physical and Life Sciences Directorate, Lawrence Livermore National Laboratory, Livermore, CA, USA. ✉e-mail: mayali1@llnl.gov; zehrj@ucsc.edu

basin scale[6,7]. Diverse diazotrophic Bacteria and Archaea have been shown to have the potential to fix $N_2$ as identified through amplification of their *nifH* genes[8], which encodes a component of the nitrogenase enzyme that catalyzes $N_2$ fixation[9]. Photoautotrophic cyanobacteria, such as *Trichodesmium*, heterocyst-forming symbionts of diatoms, and unicellular cyanobacteria (*Crocosphaera* and the symbiont UCYN-A), all have been shown to be important $N_2$-fixers in warm, low-nutrient, surface ocean waters through culture-based studies[10–13] and single cell analyses[14–17]. However, amplification of *nifH* genes from ocean waters shows that there are abundant and diverse non-cyanobacterial diazotroph (NCD) *nifH* sequences[8,18,19] that often exceed the relative abundance of amplified cyanobacterial *nifH* genes[18,20,21]. The presence of diverse *nifH* genes from NCDs suggests NCD $N_2$ fixation may be an important process in the euphotic zone, however it has not yet been directly demonstrated that these marine NCDs fix $N_2$, which is a critical first step for determining their contribution to measured community $N_2$ fixation rates.

NCDs, which could be heterotrophic or photoheterotrophic $N_2$-fixing Bacteria or Archaea, have been largely considered insignificant in biological $N_2$ fixation in marine surface waters[6]. The significance of NCD $N_2$ fixation has been questioned because of the low concentration of organic matter and relatively high concentrations of dissolved oxygen, which inactivates nitrogenase, in surface waters[6,22]. Heterotrophic $N_2$ fixation has been suggested to occur in or on particles[18,23–27] which could provide a rich source of C[18]. In particles ≥1 mm in diameter, microbial respiration-induced microaerobic zones[18,28–30] could provide suitable conditions for heterotrophic diazotrophy. In line with these hypotheses, it has recently been observed that many NCDs are motile and may use chemotaxis to locate and colonize particles suitable for $N_2$ fixation in coastal waters[31]. Additionally, NCD *nifH* genes have been found on individual particles in the North Pacific Subtropical Gyre[32]. Nitrogenase proteins from putative NCDs have been visualized associated with particles by immunolabeling in estuarine samples[33]. Despite this indirect evidence that NCD $N_2$ fixation may be occurring on particles, direct evidence of NCD $N_2$ fixation has yet to be demonstrated.

NCDs could be a significant component of marine $N_2$ fixation, but their activity and quantitative significance has yet to be directly demonstrated. The potential $N_2$ fixation rates of NCDs are difficult to assess since most open ocean NCDs do not have cultured representatives. Marine NCD $N_2$ fixation rate measurements are limited to a few cultured representatives from estuarine environments with rates of 0.02 to 1.1 fmol N $cell^{-1}$ $d^{-1}$ (scaled to per day rates assuming 24 h of $N_2$ fixation)[34–36]. Unicellular, surface ocean cyanobacteria such as UCYN-A have much higher single cell $N_2$ fixation rates of 2 to 220 fmol N $cell^{-1}$ $d^{-1}$ [15,16]. Additionally, a few indirect measurements of community $N_2$ fixation rates (0.7 to 8 nmol N $l^{-1}$ $d^{-1}$) have been inferred from locations where cyanobacteria were reportedly absent (reviewed in Moisander et al., 2017)[37]. Nanoscale secondary ion mass spectrometry (nanoSIMS) analysis has been used extensively to identify single cell activity, including $N_2$ fixation by uncultured diazotrophs[38,39]. Diazotrophs for which there are 16S rRNA gene sequences can be identified by catalyzed reporter deposition-fluorescence in situ hybridization (CARD-FISH) or related methods[14,38,40] and shown to fix $N_2$ by nanoSIMS analysis measuring cellular $^{15}N$ incorporation. However, most NCDs are only known by their *nifH* gene sequence, so visualization and identification using 16S rRNA gene-based CARD-FISH is not possible. Furthermore, FISH-nanoSIMS is generally low throughput, making surveys for potentially rare organisms, such as NCDs, impractical, due to the difficulty in identifying and mapping rare cells.

In this study, we use a dual isotope nanoSIMS approach to determine if NCDs are fixing $N_2$ in surface waters of the North Pacific Subtropical Gyre. We incubate seawater samples in $^{15}N_2$ and $^{13}C$-bicarbonate as a conservative approach to distinguish between cyanobacteria and NCDs, the former presumably fixing both $CO_2$ and $N_2$ and the latter only $N_2$. Although, this untargeted approach did not require identifying NCDs by CARD-FISH or other means, it did require a large survey of presumed cells on particles. Since it was not possible to confirm the enriched regions were cells by an independent method, cell-like regions will be described as ROIs or putative NCDs. We also measure community (bulk water) $N_2$ fixation rates using $^{15}N_2$ and determined diazotroph diversity by *nifH* gene sequencing. We present measurements of putative NCD $N_2$-fixation rates in the surface ocean.

## Results and discussion

Diazotroph diversity was investigated by sequencing *nifH* genes amplified from filtered seawater samples (Fig. 1). We detected both cyanobacterial and NCD *nifH* sequences at all locations sampled (inset map Fig. 1, Ocean Data View[41]). Three locations had comparable relative abundances of cyanobacterial and NCD *nifH* sequences (stations 5, 10 and 20), while the western-middle stations were dominated by either cyanobacterial sequences (station 14) or NCD sequences (station 17). The two eastern most stations (stations 22 and 23) had similar compositions with ~¾ of the total sequences identified as cyanobacterial and the remaining ¼ of sequences identified as NCD. The most common non-cyanobacterial *nifH* sequence was the γ-proteobacterium known as "Gamma A"[8,42], also identified as γ−247211A[43]. Relative abundances of Gamma A ranged from 34 to 99% of the total non-cyanobacterial *nifH* sequences at each location (Supplementary Fig. 1). The non-cyanobacterial sequence with the second highest relative abundance was also likely a γ-proteobacterium (*nifH* cluster 1G) and ranged from 1 to 48% of total NCD sequences per sample. Sequences from the 1O/1P cluster (namely β-proteobacteria) were also present but at much lower relative abundances (0–6% of NCD sequences). It is important to note that whole seawater *nifH* sequencing alone does not provide information about which NCDs may be particle-attached versus free-living. In fact, some particle-attached sequences may have been missed entirely as

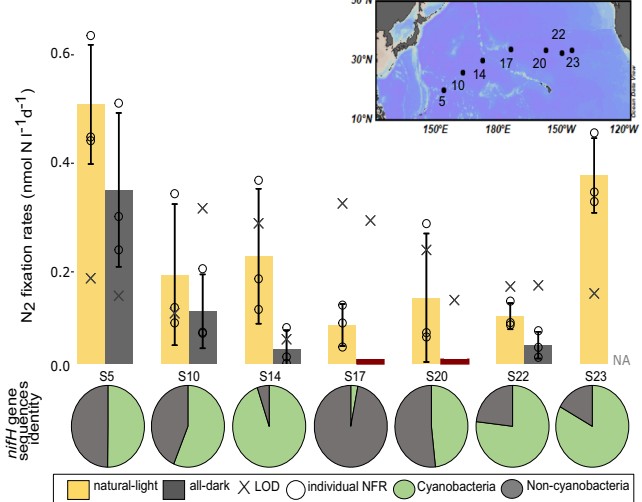

**Fig. 1 | $N_2$ fixation rates and community composition by station.** Map inset shows stations (S) sampled across the North Pacific Subtropical Gyre. Bar chart shows community $N_2$ fixation rates averages under natural-light (yellow) and all-dark (dark grey) conditions derived from biological triplicates. Individual $N_2$ fixation rates values are shown as black circles for each station and light treatment. Error bars are the standard deviations of the averages ($n = 3$). The limit of detection (LOD) for the $N_2$ fixation rates are shown with an X. Stations where the X is above the average are below the LOD but above the minimum quantifiable rate (MQR), all-dark values at S17 and S20 were below LOD and MQR (red), no data were available for S23 all-dark incubation. The lower pie charts show the relative proportion of cyanobacterial (green) and NCD (light grey) *nifH* sequences at each station. Source data are provided as a Source Data file.

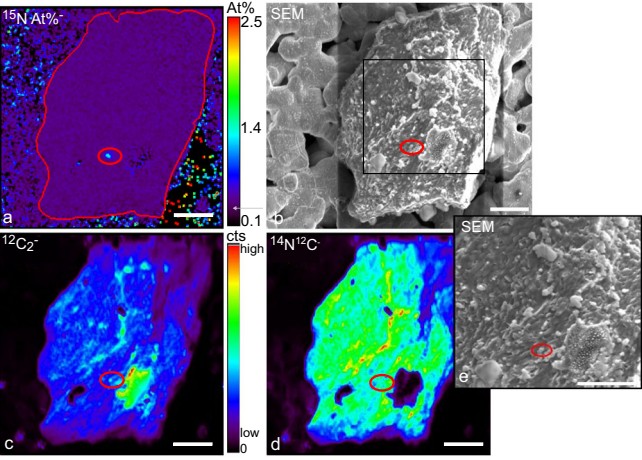

**Fig. 2 | Detailed images of one particle with an associated putative NCD (small red circle). a** $^{15}$N At% values highlight NCD ROI enriched in $^{15}$N compared to surrounding particle (within thin red outline) with enrichment near natural abundance (purple). The grey arrow on the color scale bar indicates natural abundance of $^{15}$N At %. A total of nine particles from two stations were found with one or more $^{15}$N enriched ROIs. **b** A scanning electron microscope (SEM) image of the particle and associated cells. One particle with an attached putative NCD was imaged with SEM after nanoSIMS. The black square outlines an area of closer SEM inspection shown in image **e**. Images **c** and **d** show $^{12}C_2^-$ and $^{12}C^{14}N^-$ counts (cts) collected during particle analysis with nanoSIMS. Color scale bar is for both **c** and **d**, with warmer colors representing higher counts. The cell-like ROI is visible in $^{12}C_2^-$ but obscured in d by the high overall values of $^{12}C^{14}N^-$ of the particle. Scale bar is 5 μm.

previous studies found that sequences from individual particles (>20 μm) were not well-represented in the whole water column diazotrophic community composition[26,32]. Other studies have further indicated the possibility that some diazotrophic diversity may be missed or misrepresented due to PCR biases in *nifH* amplification and sequencing[22,44]. Regardless, this analysis verifies that NCDs were a part of the diazotroph community in these samples.

To determine the activity of the diazotrophic populations at specific locations, we measured community (bulk water) $N_2$ fixation rates, which were relatively low (0.11 to 0.51 nmol N l$^{-1}$ d$^1$) (Fig. 1) compared to the well-studied Station ALOHA site in the North Pacific Subtropical Gyre (0.3 to 21 nmol N l$^{-1}$ d$^{-1}$)[3]. The natural-light $N_2$ fixation rates were all above the minimum quantifiable rate (MQR) (3 stations were >limit of detection (LOD)), as were 4 out of 6 all-dark $N_2$ fixation rates (1 station was >LOD). The all-dark $N_2$ fixation rates at stations 17 and 20 were below MQR, while the all-dark data for station 23 were not available. Quantifiable all-dark $N_2$ fixation rates accounted for 24 to 69% of the natural-light $N_2$ fixation rates. For context, all-dark $N_2$ fixation as a percent of the natural-light $N_2$ fixation in the NCD-dominant South Pacific Gyre ranges from 28% to over 100%[45] and 63% to over 100% in a temperate estuary[46].The comparison of all-dark to natural-light community $N_2$ fixation rates could be used to infer photosynthesis-independent $N_2$ fixation. However, cyanobacteria that fix $N_2$ in the dark such as *Crocosphaera*, and *Cyanothece*[47] as well as the endosymbiont (spherical body) of the diatom *Epithemia pelagica* that fixes $N_2$ in the light and dark alike[48] are all present in our samples according to *nifH* gene sequences (*Crocosphaera*: 0–39%, *Cyanothece and Epithemia pelagica* spherical body <1%, Supplementary Fig. 1) and make all-dark $N_2$ fixation rates unreliable for estimating $N_2$ fixation by NCDs. All stations in this study had both NCD and cyanobacterial *nifH* sequences present, thus preventing conclusions of NCD $N_2$ fixation without making cell- specific measurements.

Initially, our nanoSIMS analyses were untargeted, collecting data for unattached and particle-attached regions of interest (ROIs) within a given filter area. During the untargeted analysis, we found several

$^{15}$N-labeled putative NCD cells on particles but no unattached NCDs. Although we did find other unattached ROIs, including cyanobacteria-like ROIs, we do not make quantitative conclusions about the abundance of free-living NCDs because the 0.2 μm silver filters used for the analysis have micron-scale pores that may have obscured smaller unattached cells in a biased way (Fig. 2). Previous studies suggested the presence of likely unattached NCDs through small size-fractioned *nifH*-based qPCR abundances (<3.0 or 10 μm), from $3.0 \times 10^2$ *nifH* copies L$^{-1}$ in the Eastern Tropical South Pacific[22] to $3.0 \times 10^4$ *nifH* copies L$^{-1}$ in the North Pacific Subtropical Gyre[49], although particle dissociation during filtration cannot not be discounted. Furthermore, although based on a different method to estimate abundances, unattached NCD abundances (<1.6 or 3 μm) in the oligotrophic North Pacific are reported orders of magnitude higher when estimated with a primer-free metagenomic based approach at $1.3 \times 10^6$ cells L$^{-1}$[44] and flow cytometry based estimates using diazotroph relative abundance at $2.8 \times 10^6$ cell L$^{-1}$[50]. Primer free methods suggest NCDs are the dominant type of diazotroph compared to cyanobacteria in the smaller size fraction from the Tara Ocean Database[21,50].

We focused subsequent nanoSIMS analyses on particles, mapping and analyzing ~150 particles from each station. A total of 34 out of 74 $N_2$-fixing ROIs were identified as putative NCDs as they were enriched in $^{15}$N but lacked $^{13}$C enrichment, indicating no detectable $CO_2$ fixation occurred. The 34 NCD ROIs were found associated with 9 particles over 2 out of 7 stations (stations 5 and 10) with an average size of $0.8 \pm 0.3$ μm. The remaining 40 ROIs (from the 74 total) were cyanobacterial-like diazotrophs as they were enriched in both $^{15}$N and $^{13}$C (stations 5, 14, 22 and 23). All samples, including those containing active putative NCD cells, were from fully oxygenated surface waters. NCD and cyanobacterial-like ROIs were both found at stations 5 and 22, while station 10 only had NCD ROIs and stations 14 and 23 only had cyanobacterial-like ROIs. The presence of putative NCDs showed no relationship to bulk seawater nutrient concentrations (nitrate: 0.1 μM, 0.1 μM, <LOD,< LOD, 0.2 μM, <LOD,< LOD, ordered by station; phosphate: <LOD at all stations). The majority of NCD ROIs (32 putative cells) were visually particle-associated (Figs. 2a, 3a), while a small fraction of measured NCD (2 putative cells) were particle-adjacent. Particle-adjacent NCD were not directly visualized attached to particles but were found in the vicinity (<4 μm) of a particle during nanoSIMS analysis. Their proximity to a particle suggests the putative NCD may have dislodged from the particle or could have been connected to the particle by a polymeric substance that was lost during sample preparation. Conversely, it is possible diazotrophic ROIs appear associated with particles when they were actually free-living by randomly landing on particles or being forced into soft particles during filtration. Further studies are needed to validate the presence of NCDs on suspended marine particles. Regardless of whether particle associated or not, the data show putative NCDs are capable of $N_2$ fixation in the oxygenated surface ocean.

Particles available for analysis ranged from 5 to 210 μm in diameter (pore size of the incubation pre-filter) and included densely packed particles (Fig. 2, Supplementary Fig. 2) as well as loosely formed aggregates (Supplementary Fig. 2). The particle compositions and internal nutrient concentrations are unknown although the relatively high ion signals of both $^{12}C_2^-$ and $^{12}C^{14}N^-$ measured by nanoSIMS show the particles have a high organic content (Fig. 2c, d). We do not know if NCDs would have been found on larger particles that could provide microaerophilic zones conducive to $N_2$ fixation. Further, we did not measure the size of the particles that exceeded the size of the nanoSIMS analysis areas (20 to 40 μm), so some of the analyzed particles were likely larger. We found NCD ROIs on a wide range of particle size from 6 μm to >20 μm. Particles with one or more associated NCD ROI accounted for 1.5% of the total particles analyzed at station 5 and 4.1% at station 10. Approximately 1/3 of the particles with an associated NCD ROI contained a single NCD ROI, although this is likely an

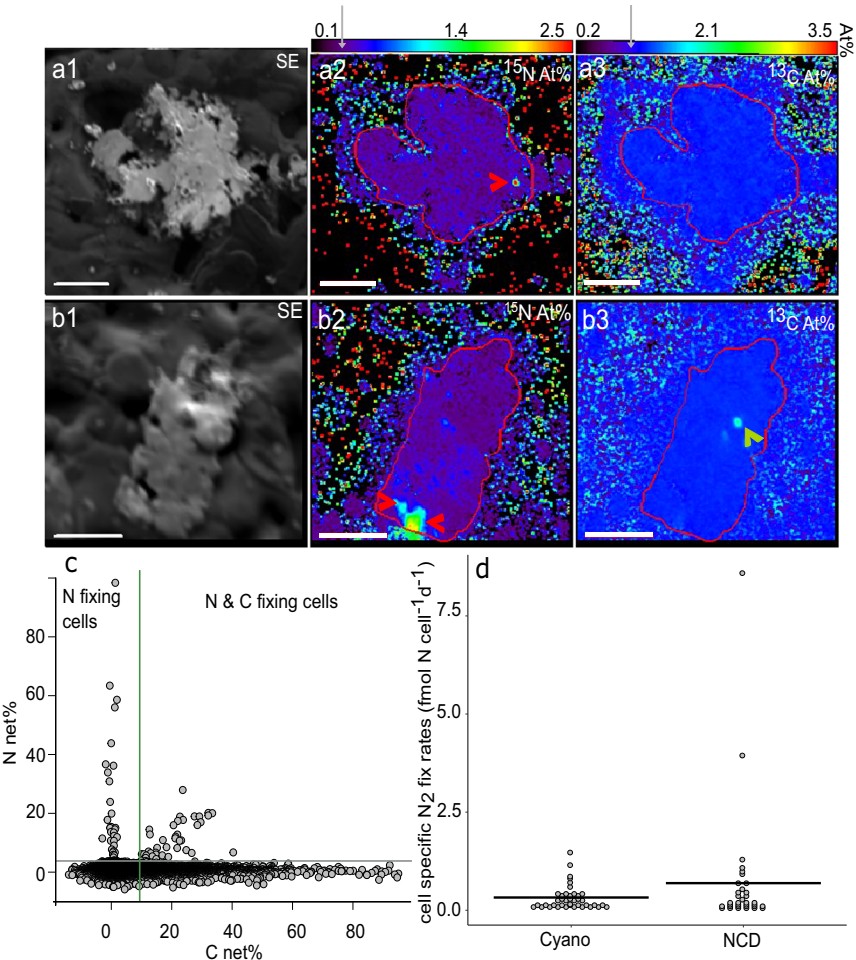

**Fig. 3 | NanoSIMS data. a, b** Example images of two particles with associated NCD ROIs. Images show particles using secondary electrons (a1, b1) with attached ROIs enriched in $^{15}N$ (red arrows, a2 and b2) but not enriched in $^{13}C$ (a3 and b3). A total of 32 NCD-like ROIs were found associated with 9 particles. Red lines outline the particle shape based on secondary electron (SE) image. Color scale bars correspond to At% enrichment for the given isotope, grey arrows on color scale bar indicate natural abundance of rare isotope. The green arrow shows an ROI enriched in $^{13}C$ but not $^{15}N$, similar to what we would expect for non-diazotrophic autotrophs (b3). Over 5700 non-diazotrophic autotroph-like ROIs were found across 7 stations. Scale bar is 5 μm. **c** Cnet% versus Nnet% of ROIs ($n = 39,043$), lines show Cnet% and

Nnet% minimum enrichment thresholds. The minimum enrichment thresholds section the data into quadrants. Data points in the upper left quadrant are ROIs enriched in $^{15}N$ and counted as NCDs ($n = 34$). Points in the upper right quadrant are ROIs enriched in $^{13}C$ and $^{15}N$ and classified as cyanobacterial-like ROIs ($n = 40$). The lower two quadrants constitute heterotrophic ROIs (bottom left, $n > 33,245$) and C-fixing (autotrophic, $n > 5724$) ROIs (bottom left) which make up 99.8% of the data analyzed. **d** Putative cell-specific $N_2$ fixation rates for ROIs enriched in $^{15}N$ from all stations grouped as NCD-like ROIs ($n = 34$) and cyanobacterial-like ROIs ($n = 40$). Data points represent individual ROIs and the horizonal line shows the mean for each group. Source data are provided as a Source Data file for **c, d**.

underestimate as we did not acquire data through entire particles (nanoSIMS is a surface analysis technique). To examine this further, we randomly picked four particles with an associated NCD ROI and analyzed these particles at multiple depths (Supplementary Fig. 3), finding additional NCD ROIs in 3 of the 4 particles. As such, putative NCD occurrences are underestimated due to missed $N_2$-fixing ROIs that were within or on the underside of the analyzed particles.

We estimated the percent of new biomass synthesized from fixed N and C relative to total biomass, Nnet% and Cnet%, respectively[51,52], for each $N_2$-fixing putative cell. The cell enrichment minimum thresholds (3x the standard deviation of unlabeled *Pseudomonas* cells) for both $^{15}N$ and $^{13}C$ correspond to Nnet% = 3.1% and Cnet% = 9.4% (Fig. 3b). Putative NCD cells exhibited a broad range of Nnet% from 4.02 to 98%, while Cnet% ranged from −2.9 to 1.9%. While it is possible the new N biomass synthesized by putative NCDs may be due to secondary uptake of $^{15}N$ labeled ammonium ($NH_4^+$) derived from cyanobacteria, it is unlikely those values would be high enough to pass the conservative minimum $^{15}N$ threshold (0.47 At%) used to define an $N_2$-fixing ROI given the initial amount of $^{15}N$ added to the incubation (2.8 At% ± 0.4).

For context, the minimum $^{15}N$ threshold is at least 1.25x higher than the average uptake by plankton of diazotroph derived N during a cyanobacterial bloom in 24 h[53–55]. Additionally, the few high $^{15}N$ enrichment values measured in this study are highly improbable to have resulted from secondary uptake during the 24 h incubation. The Nnet% of cyanobacterial-like ROIs ranged from 3.8 to 27.9%, while Cnet% had a minimum of 9.9% and a maximum of 40.4%. For the cyanobacterial-like ROIs, Nnet% and Cnet% were not proportional, indicating the putative cells were obtaining either their N or C from other unlabeled sources. For the putative NCDs, 4 ROIs exhibited Nnet% values of 50% or higher, indicating that based on N, half or more of the cellular biomass had been synthesized during the isotope incubation. At this rate, NCDs could fulfill their cellular N requirements through $N_2$ fixation in 24 h. The highest observed Nnet%, 98% (measured in one putative cell), is equivalent to 4 cell divisions in 24 h. Such high growth rates have been observed in NCD cultures under nutrient replete conditions, but this is unusually fast compared to the average generation time of marine planktonic and particle attached bacteria (9–20 h)[46,56,57] and may therefore reflect a smaller number of asymmetric divisions such as

occurs in Planctomycetes[44,58]. A Nnet% <50% could indicate a growth rate slower than the incubation period (24 h) or that the cells use other sources of N in addition to $N_2$.

In addition to new biomass, we also calculated cell-specific $N_2$ fixation rates for each particle-associated NCD ROI to quantify net $N_2$ fixed over the duration of the $^{15}N_2$ incubation by individual putative cells. Putative cell-specific rates ranged from 0.05 to 8.61 fmol N cell$^{-1}$ d$^{-1}$ (Fig. 3c) with an overall average of $0.69 \pm 1.57$ fmol N cell$^{-1}$ d$^{-1}$. The cell-specific average was higher than those measured for NCD isolates from the Baltic Sea with a maximum of 0.06 fmol N cell$^{-1}$ d$^{-1}$ (scaled up to 24 h of $N_2$ fixation) in minimal culture media and up to 0.2 fmol N cell$^{-1}$ d$^{-1}$ with replete labile C and $NH_4^{+}$[46], which could be present at high concentrations on marine particles. However, the putative cell-specific average was similar to a microaerobic NCD isolate from oxygenated Baltic Sea water with a cell-specific rate of 1.1 fmol N cell$^{-1}$ d$^{-1}$[35] and an isolate from an oxygen minimum zone in the South East Atlantic with a cell-specific rate of 0.50 fmol N cell$^{-1}$ d$^{-1}$[36]. The putative NCD cell-specific rates show particle attachment may allow NCDs to fix $N_2$ in the oxygenated surface ocean contributing to the total fixed N available, a process that was previously only hypothesized, but not demonstrated, to occur in the oligotrophic ocean.

We used the single cell nanoSIMS data to estimate the contribution of putative NCDs to the total $N_2$ fixation in our samples (Summarized in Table 1). Our volumetric estimates of putative NCD abundance ranged from $3.48 \times 10^3$ to $2.9 \times 10^4$ cells L$^{-1}$, which is within the range of previously reported abundances of Gamma A, a commonly occurring NCD *nifH* sequence[49,59] and the most relatively abundant NCD *nifH* sequences recovered from our samples. Scaling the cell-specific rate averages from each station using the NCD abundance can provide a rough minimum estimate of NCD contribution to community $N_2$ fixation rates, noting that our calculations are based on underestimations. NCD $N_2$ fixation totals were up to 0.01 nmol N L$^{-1}$ d$^{-1}$, accounting for up to 4.8% of the total community $N_2$ fixation rates in the natural light. We note that if the NCD activity represented the entire community $N_2$ fixation rate, in the hypothetical absence of cyanobacteria, the NCD contribution values would fall below the MQR of the bulk measurements. Yet, if particle-attached NCD $N_2$ fixation is light independent, these low putative cell-specific rates and abundances may be significant when depth-integrated, considering the large expanse of the dark ocean[37,60–62].

The calculated contributions of putative NCD activity to total fixed N discussed above are based on analyses of only several thousands of putative cells from seven stations on one cruise and a number of assumptions, and therefore are only rough estimates of the

potential importance of this underexplored phenomenon in the world's oceans. Our underestimation derives from NCD and cyanobacterial abundances when scaling active $N_2$-fixing cell-like ROI numbers to volumetric abundances (cells L$^{-1}$). The areas scanned during nanoSIMS analysis are missing $N_2$-fixing ROIs that were in the interior or underside of particles, lowering the estimated cell abundances. Evidence of this is demonstrated by particle depth profiles (Supplementary Fig. 3) in which 3 out of 4 particles harbored additional NCD ROIs within the interior volume of the particle that were not measurable from the surface of the particle. To amend our estimates, we calculated NCD ROIs per unit volume using an average number of 3 NCD ROIs per particle, which takes into account the average NCD ROIs found in the depth profiles. Additionally, Niskin bottles are likely to under-sample particles, especially fast-sinking ones[63], while use of the 210 μm prefilter further reduced the particle concentrations used to estimate volumetric NCD abundances. Our putative cell-specific $N_2$ fixation rates may further be underestimated due to dilution of the $^{15}N$ (also impacting the corresponding $^{13}C$ dilution for cyanobacteria) caused by the fixation process determined by Meyer et al., 2021[64] of up to 12% dilution (not included in our calculations). The comparison of bulk $N_2$ fixation rates to community contribution from single cell nanoSIMS analysis also supports that our calculations were underestimates, since the bulk rates were greater than the total volume of integrated NCD and cyanobacterial single cell data (maximum 24% of the bulk). Regardless of these underestimations, our empirical measurements still show that putative NCD activity can account for a portion of the community $N_2$ fixation rates, contributing to the fixed N in the system.

We show evidence of putative NCDs fixing $N_2$ in the surface ocean, and nearly all encountered NCD ROIs were associated with particles. It has been hypothesized that particles may provide favorable microenvironments for NCD because they may have low oxygen[25,29,65], high C concentrations, and beneficial C:N ratios[18,66]. For heterotrophic NCDs, particle attachment would provide a readily available source of organic carbon compared to the water column. However, microaerophilic environments may not be a factor for the small particles surveyed in this study, as estimates for particle size ranges needed to harbor a microanoxic zone are ≥0.6 mm in diameter[30], and we found active putative NCDs on particles as small as 6 μm diameter. If microanoxic zones are required for $N_2$ fixation, it is possible small particles with associated NCD may have initially been large enough to harbor microaerobic zones but were fractured during filtration or analysis preparation.

We found diazotrophic cyanobacteria-like ROIs at 4 out of the 7 stations yet the overall number of cyanobacteria-like ROIs encountered was similar to NCD ROIs (40:34) from 2 stations. It is possible the

**Table 1 | Summary of community $N_2$ fixation rates and nanoSIMS calculations by station**

| station | natural-light community $N_2$ fixation rate (nmol N l$^{-1}$ d$^{-1}$) | all-dark community $N_2$ fixation rate (nmol N l$^{-1}$ d$^{-1}$) | particles l$^{-1}$ x 10$^4$ (5–150 μm) | total particles analyzed by nanoSIMS | % particles found with NCD | # NCD cells per station | NCD $N_2$ fixation rates (fmol N cell$^{-1}$ d$^{-1}$) | NCD abundance (cell L$^{-1}$) | NCD $N_2$ fixation rate contribution (nmol N L$^{-1}$ d$^{-1}$) | % NCD $N_2$ fix to community light $N_2$ fix rates | % NCD $N_2$ fix to community dark $N_2$ fix rates |
|---|---|---|---|---|---|---|---|---|---|---|---|
| 5 | 0.51 ± 0.11 | 0.35 ± 0.14 | 7.57 (± 1.07) | 196 | 1.50 | 5 | 2.92 (±3.49) | 3.48 × 10$^3$ | 0.01 | 1.99 | 2.9 |
| 10 | 0.19 ± 0.13 | 0.12 ± 0.06 (x) | 20.4 (± 8.98) | 146 | 4.11 | 29 | 0.31 (±0.33) | 2.94 × 10$^4$ | 0.01 | 4.80 | 7.6 |
| 14 | 0.23 ± 0.12 (x) | 0.06 ± 0.03 (x) | 7.79 (± 4.30) | 123 | – | 0 | – | – | – | – | – |
| 17 | 0.10 ± 0.03 (x) | <MQR | 3.21 (± 0.84) | 217 | – | 0 | – | – | – | – | – |
| 20 | 0.15 ± 0.11 (x) | <MQR | 7.22 (± 1.22) | 182 | – | 0 | – | – | – | – | – |
| 22 | 0.12 ± 0.02 (x) | 0.06 ± 0.02 (x) | 3.28 (± 0.31) | 126 | – | 0 | – | – | – | – | – |
| 23 | 0.38 ± 0.07 | NA | 4.63 (± 1.63) | 115 | – | 0 | – | – | – | – | NA |

Calculations represent values scaled-up from a limited analysis and should be considered as rough estimates included for context. The basis of the estimates were made using the particle concentration and the percent of analyzed particles with an associated putative NCD to calculate 'NCD abundance', 'NCD $N_2$ fixation rate contribution', and '% NCD $N_2$ fix to community light/dark $N_2$ fix rates'. Values represent mean (± standard deviation of the mean) at each station, $N_2$ fixation rates marked with an 'x' are above the MQR but below the LOD.

particle-focused analysis contributed to the lower number of cyanobacterial ROIs found as we may not necessarily expect to find cyanobacteria associated with particles. However, a study comparing *nifH* sequences from water column to particles at 150 m found that while the water column had a higher relative abundance of cyanobacteria, the sequences recovered from particles were 44.5% cyanobacterial (*Crocosphaera*, UCYN-A *Richelia* and *Trichodesmium*)[32]. A metagenomic study also showed evidence of cyanobacterial particle association based on size fractionated samples[50]. Additionally, cyanobacterial *nifH* sequences were found on particles at 4000 m (*Crocosphaera*, *Richelia* and *Epithemia pelagica*-spherical body)[48,67] yet the cyanobacterial diazotrophic activity on particles is unknown. As such, the cyanobacterial abundances could be due to typically C-fixing cyanobacteria using heterotrophic C acquisition while attached to particles and would therefore be categorized as a NCD in this study. Both *Trichodesmium* and *Cyanothece* have shown evidence of mixotrophy by incorporating dissolved organic C while maintaining their $N_2$-fixing capability[68,69]. However, the very low Cnet% of the ROIs classified as NCDs indicate the ROIs are likely heterotrophic, not mixotrophic and did not fix any C at 15 m depth light levels over the 24 h incubation. Nonetheless, future studies on particle associated diazotrophs that can incorporate cell identity as well as C and $N_2$ fixation rates would lead to valuable insights into the C-fixing potential of particle attached cyanobacteria.

This study identified putative NCDs associated with organic particles. These putative NCDs are identified based on organic matter puncta that are enriched in $^{15}N$ derived from the provided $^{15}N_2$ gas but not enriched in $^{13}C$ from the provided $^{13}C$-bicarbonate. Our approach did not enable us to directly link pre-identified cells, such as those visualized with DAPI staining, with the $^{15}N$ enriched ROIs from our nanoSIMS measurements. Although we cannot definitively identify the $^{15}N$ enriched areas as cells, to the best of our knowledge the probability that they are diazotrophs is high as only $N_2$-fixing organisms could convert $^{15}N_2$ gas into $^{15}N$ enriched biomass, especially in a cell-like size and shape. However, in the absence of such a direct link, we call these putative NCDs to allow for alternative possibilities that could potentially result in a $^{15}N$ enrichment, yet are unlikely, as discussed below. If the $^{15}N$ enriched ROIs are not complete cells but a partial or a lysed cell it would still indicate $N_2$ fixation by NCDs as the biomass is enriched in $^{15}N$. Additionally, the added isotope is gaseous therefore any $^{15}N_2$ not incorporated into biomass would quickly dissipate once filtered. Alternatively, any possible unknown particle contaminant from the $^{15}N_2$ gas that would result in a cell-like shape would have an At% similar to that of the original gas (98 At%), rather than the <2 At% found in our ROIs. Lastly, if the $^{15}N_2$ gas was contaminated with $^{15}N$-nitrate, $^{15}N$-nitrite and/or $^{15}N$-ammonium, as has been reported previously[70], we would expect to see higher overall community $N_2$ fixation rates and a higher proportion of $^{15}N$ enriched ROIs since these N-substrates are readily taken up by many marine microbes. Prior to this study NCDs had not been visualized on particles from the open ocean, so in order to collect sufficient data it was necessary to analyze >1100 particles and tens of thousands of cells with the nanoSIMS for many weeks, using automated analysis to detect the rare NCD-like ROIs. This amount of analysis time is not typically possible in most nanoSIMS studies. The vast amount of data collected using this method precluded prior visualization of cells as the 1100 particles would have needed to be imaged and mapped in both fluorescent microscopy and nanoSIMS to encounter the 34 NCD-like ROIs found in this study. Our findings underscore that future studies should focus on NCD containing particles by identifying NCDs, possibly using either nitrogenase immunolabeling techniques[33] or a *nifH* gene targeted in situ hybridization approach such as geneFISH[71], prior to nanoSIMS measurements. Although we lack the definitive connection between cells and ROIs, our study is an informative step toward showing NCD-like ROIs that do not incorporate $^{13}C$-bicarbonate but do incorporate $^{15}N_2$ are present on

particles in the surface ocean and that at least some of the NCDs in the surface ocean are fixing $N_2$.

Our results directly showing $N_2$ fixation by putative NCDs attached to particles in surface waters of the oligotrophic ocean sheds light on how these organisms may overcome the difficulties of heterotrophically-fueled $N_2$ fixation in this low nutrient environment. Additionally, NCDs attached to particles could sink out of the euphotic zone extending the depth range of $N_2$ fixation. The particle-associated $N_2$ fixation would have local effects on particle dynamics, allowing for C remineralization even after initial bioactive N sources were depleted[26,30]. Our data showing previously unidentified particle-associated heterotrophic $N_2$ fixation in the surface ocean demonstrates the need for more careful evaluation of microscale interactions in these processes.

## Methods

Seven locations (Fig. 1) spanning the North Pacific Subtropical Gyre between Guam and San Francisco (November 2019) were analyzed for the presence of $N_2$-fixing organisms by *nifH* sequencing, community $N_2$ fixation rate measurements and putative cell-specific activities of NCDs.

### Sample collection
Samples were collected using Niskin bottles attached to a CTD profiler from surface seawater (15 m). All water samples were collected in acid-cleaned polycarbonate bottles rinsed 3 times with local seawater. Large grazers were removed while the bottles were filled using a 210 μm Nitex™ plankton netting (BioQuip, Rancho Dominguez, CA). Samples collected from each station include: natural isotope abundance (triplicate samples of 2 L), nutrient analysis (replicates of .05 L), DNA (replicates of 2 L) and isotope incubations (triplicates of 4.4 L). NanoSIMS samples (0.1–0.5 L) were subsampled from isotope incubations. Bottles for isotope incubations were immediately placed in surface seawater flow-through incubator shaded to ~15 m light intensities until isotope addition (~1 h after initial sampling). Natural isotope abundance and DNA water samples were filtered immediately. Nutrient concentrations (nitrate + nitrite and phosphate) were analyzed according to EPA protocol (40 CFR part 136, appendix B), and the method limit of detections were 0.01 μM and 0.02 μM, respectively.

### DNA extraction and *nifH* sequencing
Diazotroph diversity was assessed by PCR amplification and sequencing of the *nifH* gene. Surface seawater (2 L, 15 m, <210 μm) was filtered onto 0.2 μm filters (PE 25 mm; Supor-200; Pall Life Sciences, Port Washington, NY, USA) using peristaltic pumps. Filters were flash frozen before storage at −80 °C. DNA was extracted with DNeasy Plant Mini Kit (Qiagen, Hilden, Germany) with modifications for increased cell lysis[43]. The *nifH* gene was amplified by PCR using the universal *nifH* outer primers, YANNI/450, and common-sequencer linkers with inner primers, up/down, in a nested reaction[72]. Amplicons were sequenced with Illumina MiSeq sequencing (2 × 300 bp, with a targeted sequencing depth of 20,000 per sample) at the University of Illinois Genome Research Core Facility. Raw sequences were processed as described in Cabello et al., 2020[73]. Briefly, sequences were quality controlled and clustered at 97% nucleotide identity using Qiime[74], representative sequences were assigned phylogeny with BLASTX and operational taxonomic unit tables were rarified according to the sample with the lowest sequence recovery (738 sequences).

### $^{15}N_2$ incubations
To investigate the $N_2$ fixation activity of NCDs, triplicate 15 m depth seawater samples (<210 um, 4.4 L) were injected with a $^{15}N_2$ gas bubble (98%+, lot # - I-22779, Cambridge Isotopes, Tewksbury, MA, USA) following the bubble release method[75]. Bottles were rolled back and forth to equilibrate the $^{15}N_2$ bubble for 20 minutes (~25 rpm) before bubble

release and $^{13}$C-bicarbonate addition (60 μM, 99%, Cambridge). Triplicate incubations were transferred back to the incubators shaded to light intensity at 15 m depth (7 stations) or complete darkness (six stations) for 24 h.

## Community N$_2$ fixation rates

Isotope incubations were used for particulate organic nitrogen measurements for community N$_2$ fixation rates. Membrane inlet mass spectrometry (MIMS) samples were taken at the end of 24 h incubations from each bottle to measure the dissolved $^{15}$N$_2$-gas enrichment available (2.8 At% ± 0.4). Incubation bottles were opened, and MIMS samples were siphoned into glass vials (15 mL) which were then stoppered and crimped with aluminum caps. Samples were kept at 4°C until measured. After subsampling for nanoSIMS samples, biological triplicates were vacuum filtered (3.5–3.8 L) through pre-combusted (4 h at 450°C) 25 mm GF/F (Whatman®) and flash frozen before storage at −80°C. Additional water samples (2 L) were collected from each location in triplicate and immediately filtered with a separate vacuum filtration system for $^{15}$N natural abundance measurements, except for station 14 which only had duplicate samples available. All filters were dried at 75°C for 72 h before being pelleted in tin foil disks (30 mm, Elemental Analysis, Okehampton, UK). The $^{15}$N enrichment and natural abundance of particulate organic N was measured along with blanks using a Carlo-Erba EA NC2500 coupled with Thermo Finnigan Delta-Plus XP at the University of Hawaii Stable Isotope Facility. N$_2$ fixation rates as well as the MQR (lowest rate quantified with confidence provided the propagated errors for replicate samples) and LOD (lowest value considered detected, final At% − At% of natural abundance = 0.00146 At%) were calculated as in Gradoville et al., 2017[76] according to Montoya (1996)[77]; all values are listed in Table 1. Rate values above LOD as well as values that fell below the LOD but above the MQR are reported. Community N$_2$ fixation incubation protocols and data reporting follow recommendations from White et al., (2020)[78]: all samples were run in triplicate except as described above (T$_o$ station 14) and $^{15}$N incubation filters contained >10 μg N per filter. Our experimental design differed from recommendations by White et al., (2020)[78] in that incubations were not initiated before dawn, our 24-h incubations started at various times throughout the day.

## NanoSIMS analyses

Subsamples of the isotope incubations were filtered (0.1–0.45 L) through 0.2 μm pore-size silver membrane filters (25 mm, Cole-Parmer, Vernon Hills, IL, USA) then fixed (1.8% formaldehyde) at room temperature for 2 h. Filters were flash frozen before storage at −80°C until nanoSIMS analysis (CAMECA NanoSIMS 50, Lawrence Livermore National Laboratory). NanoSIMS measurements were made with a -2 pA Cs$^+$ primary beam, and data were collected for the masses of $^{12}$C$_2^-$, $^{12}$C$^{13}$C$^-$, $^{12}$C$^{14}$N$^-$ and $^{12}$C$^{15}$N$^-$ where $^{12}$C$^{13}$C$^-$/$^{12}$C$_2^-$ = 2 · $^{13}$C/$^{12}$C; $^{12}$C$^{15}$N$^-$/$^{12}$C$^{14}$N$^-$ = $^{15}$N/$^{14}$N[79]. Data were collected with a raster of 20 × 20 to 40 × 40 μm$^2$ with 256 × 256 pixels and 30–60 cycles for each analyzed area. Before each analysis, the area was automatically sputtered to a depth of ~60 nm to establish sputtering equilibrium, and the secondary ion beam and mass peaks were automatically centered. Each analysis took 35–65 min of instrument time. NanoSIMS measurements were conducted with both untargeted and particle-targeted approaches. The untargeted approach used an automated analysis routine which sequentially moved the stage to scan non-overlapping raster areas to map contiguous areas of the filter, and at each new location an automated peak centering was redone before data collection. To target particles, we first mapped a large number of them using nanoSIMS secondary electron imaging and then we used the same automated analysis routine to target particle locations. These analyses only included the particle surface. Several of the targeted particles were subsequently analyzed at

multiple depths by serially eroding into the particle with a high Cs$^+$ current followed by data collection. Images were processed using L'Image software (developed by L. Nittler, Carnegie Institution of Washington, Washington, DC, USA) where they were aligned, and drift corrected. Cell-like regions were identified as regions of interest (ROIs) based on a threshold of 150 counts of $^{12}$C$^{14}$N$^-$ per pixel. The data for these ROIs were extracted and then quality controlled by removing data for ROIs with <100 total counts for at least one of the minor isotopes or error >30% of the ratio value. These thresholds were set to allow for small cells. Out of the 1000 s of ROIs analyzed it is likely some percent are not cells but other organic matter from the complex organic particles they are associated with, therefore these are referred to as ROIs rather than cells. Cell-like ROIs were considered to have incorporated a statistically significant amount of an isotopically labeled substrate if the isotope enrichment was greater than that of unlabeled reference cells (*Pseudomonas*) by 3x the error associated with the minimum acceptable count for an ROI; these thresholds correspond to 1.23 atom percent (At%) $^{13}$C and 0.47 At % $^{15}$N, which are relatively conservative. We used our *Pseudomonas* reference cells because they provide a consistent amount of biomass and their isotopic composition was statistically indistinguishable from the unlabeled environmental sample ROIs measured with nanoSIMS in this study. Additionally, we only counted putative NCD cells as enriched in $^{15}$N if they had clear cellular outlines in $^{12}$C$_2^-$, $^{12}$C$^{13}$C$^-$, $^{12}$C$^{14}$N$^-$ or $^{12}$C$^{15}$N$^-$ and the enrichments were distributed throughout the entire ROI.

Enrichment values of $^{13}$C and $^{15}$N were used to calculate net assimilation percent (Xnet%, where X can be C or N), which we use to estimate newly synthesized biomass relative to total biomass assuming no change in cell stoichiometry. For N, Nnet% = [F$_s$/(F$_s$ + F$_i$)] *100, where F$_s$ is the N fraction derived from the isotope substrate and F$_i$ is the N fraction of the original biomass[51]. F$_s$ and F$_i$ are further defined by Popa et al. (2007)[52] in terms of isotopic ratio of the initial pool (R$_i$), spiked pool (R$_s$) and final ratio in the cell of interest (R$_f$), where Xnet % = Fx$_{net}$%/(Fx$_{net}$%+100)

$$FxNet = \frac{Rf\left(1 - \frac{Ri}{Ri+1}\right) - \left(\frac{Ri}{Ri+1}\right)}{\left(\frac{Rs}{Rs+1}\right) - Rf\left(1 - \left(\frac{Rs}{Rs+1}\right)\right)} *100 \qquad (1)$$

Cell-specific N$_2$ fixation rates (fmol N cell$^{-1}$ d$^{-1}$) were calculated for putative cells that were above the minimum enrichment threshold (>0.47 At% $^{15}$N, as described above) following Krupke et al. (2015)[80]. The average initial bulk isotopic composition of the unlabeled ("natural abundance") samples from each location were subtracted from the At% $^{15}$N values of enriched ROIs to estimate rare isotope incorporation, and MIMS values were used to define the amount of $^{15}$N available in the enrichment. The carbon content per ROI was based on a spherical cell volume from the defined ROI using the C content per cell according to Verity et al. (1992)[81] for cells >0.6 um$^3$. Estimates for N content per cell were adjusted for heterotrophic cells by using an average C:N ratio measured in cultured heterotrophic bacterial cells of 5.2[82] and 6.3 for cyanobacterial-like cells[15]. N content for ROIs with spherical cell volume ≤ 0.6 um$^3$ were estimated according to Khachikyan et al. (2019)[83] as small cells can have proportionally higher N content than larger cells.

## Estimating putative NCD contribution from single cell measurements

We carried out calculations to scale the nanoSIMS data to a volumetric context (summarized in Table 1) and provide an approximation of the potential contribution of NCD diazotrophy to total N$_2$ fixation activity at the stations sampled. These calculations combined single cell nanoSIMS data, the frequency of particles on which we found at least one NCD ROI, and particle abundances from each station. We assumed

that when a putative NCD was present on the surface of particles, it was representative of the total NCD particles present in a sample and multiplied the number of particles by the average number of putative NCDs found per particle (3) and the average $N_2$ fixation rate of the NCD ROI. Analysis. This calculation clearly underestimates the abundance of NCDs as all particles with particle interior NCDs would be excluded and our collection approach generally under-sampled particles (discussed below). Particles ($>5\,\mu m$) were quantified by staining polycarbonate filter pieces corresponding to nanoSIMS samples with DAPI (4',6-diamidino-2-phenylindole,$1\,\mu g\,\mu l^{-1}$) and counting with microscopy (Zeiss Axioplan epifluorescence microscope) along three transects of the filter piece to calculate particle concentration (particles $L^{-1}$) A subsample of the analyzed particles was imaged with an FEI Quanta model 450 scanning electron microscope with a 5 kV beam.

All analysis after export from L'image was conducted in Excel (2209)[84] and R(4.2.1)[84] and figures were produced using R or package ggplot2 (3.3.6)[85].

### Reporting summary

Further information on research design is available in the Nature Portfolio Reporting Summary linked to this article.

### Data availability

All data is provided in manuscript and supplemental materials, with the exception of the raw *nifH* sequences, which were deposited in the Sequence Read Archive at National Center for Biotechnology Information (http://www.ncbi.nlm.nih.gov/sra) under Bioproject ID PRJNA730862. Source data are provided with this paper as a Source Data file. Source data are provided with this paper.

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

## Acknowledgements

We gratefully acknowledge Ate Visser for providing access to the MIMS and for helpful discussion, Fangjuan Huang for shipboard assistance, Colin Carney for providing EA-IMRS consultation, Rosie Gradoville for $^{15}N$ incubation and calculation discussions and Jonathan Magasin for statistical advice. We thank William Cochlan and Christopher Ikeda at San Francisco State University for nutrient analyses. We greatly appreciate the captain and crew of the R/V Sally Ride for field logistical support and Lasse Riemann (University of Copenhagen) for helpful discussions. This research was funded by the Simons Foundation (SCOPE Award ID 724220, J.P.Z.) and NSF grants (OCE-1559165 to J.P.Z.), the Lawrence Livermore Graduate Research Scholars Programs, and the DOE-BER Genomics Sciences LLNL biofuels Science Focus Area grant #SCW1039. K. Turk-Kubo is partially supported by NSF (OCE-2023498). Ship time was granted by UC Ship Funds Program supported by UC San Diego, Scripps Institution of Oceanography. Work at LLNL was performed under the auspices of the US Department of Energy at Lawrence Livermore National Laboratory under Contract DE-AC52- 07NA27344.

## Author contributions

K.J.H. and X.M. conceived and designed the study. K.J.H., E.W.K.M. and K.A.T.K. collected, incubated, and processed samples. K.J.H., K.A.T.K., X.M. and P.K.W. analyzed data. K.J.H. and J.P.Z. wrote the initial drafts with revision from all authors.

## Competing interests

The authors declare no competing interests.
