## [Peer Review File · Nature Communications]

Direct cell-specific measurements show putative N₂ fixation by particle-attached non-cyanobacterial diazotrophs in the North Pacific Subtropical GyreREVIEWER COMMENTS

Reviewer #1 (Remarks to the Author):

Review of Direct cell-specific measurements show N₂ fixation by particle-attached non-cyanobacterial diazotrophs in the North Pacific Subtropical Gyre by Harding et al.

In this study water samples collected from the North Pacific Subtropical Gyre, from seven different locations, were exposed to stable isotope dual labeling followed by single-cell and bulk isotopic measurements with the aim of quantifying nitrogen fixation by non-cyanobacterial diazotrophs (NCDs). Using dual labeling ¹⁵N₂ and ¹³C-bicarbonate, the authors differentiated between cyanobacterial (enriched in both ¹⁵N and ¹³C) and non-cyanobacterial (enriched in ¹⁵N only) diazotrophs during single-cell nanoSIMS analyses, confidently reporting single-cell N₂ fixation rates for this second group of microorganisms. The single-cell analyses were complemented by community analyses using EA-IRMS (bulk isotope measurements) and nifH gene sequencing.

The study aim is sound and clear. The experimental design including numerous replicates and controls, covering a high number of sampling sites, shows a comprehensive approach and thoughtful, well planned experimental and analytical scheme. The single-cell measurement effort is appreciable as it requires extensive nanoSIMS analytical time and operational effort. The results are openly and efficiently presented. However, in this reviewer's opinion, the results do not fully support the discussion and interpretation of the data. My main concerns are that (1) evidence that the NCDs are actually associated/attached to particles is not obvious, and (2) I find it unconvincing that ¹⁵N enriched hot spots identified and measured under the nanoSIMS are NCDs cells or microbial cells at all.

I believe the study is extremely valuable but is currently incomplete. As much as I understand the difficulties associated with such single-cell analyses, a simple microscopy picture of fluorescence in situ hybridization on one single particle using 16S or 23S rRNA specific nucleotide probe for NCD group to show its association with the particles will provide the direct proof and identification of this group. Such approach, even without being correlated with nanoSIMS analysis of the same particle (which is very difficult to achieve) will still provide a valuable evidence of the presence and association of NCDs with the targeted particles. Without such direct proof (using the suggestion above, or other means to prove that the hot spots indeed represent cells), measurements of ¹⁵N hot spots alone cannot be used to present single-cell measurements and build up discussions around single-cell NCDs rates and their importance in the oceanic N-cycling.

Specific comments:

Lines 163-165: main concern regarding cell identification, ¹²C¹⁴N- secondary ion map is not enough to identify microbial cells inhabiting an organic rich particle containing also ¹²C¹⁴N molecular ions. Was there a ³¹P map recorded as proxy for cell nucleoid?

Line 252; 267: can be that due to large volume filtration on relative small surface of the filter that only NCDs that could stick to the particles would stay on the filters and the other were pushed to the filters pores or removed during processing, e.g. by flush freezing.

What is the average cell size of a NCDs cell?

Line 276 and line 333: Where is the proof of NCDs cell identification? Can the author rule out the ¹⁵N enrichment in the cell was not transferred from a cyanobacterial diazotroph within the 24h time of isotope labeling? Can the authors prove that the ¹⁵N hot spots in the particle are microbial cells at all?

Line 282: "The majority of NCDs (66 cells) were clearly particle-attached.."

Besides the identification issue pointed out above, as the water samples were deposited

on the filter by filtration of a large volume of 1 L or higher how one can claim that cells on the particle surface are particle attached and not accidentally attached during filtration?

What is the composition of the particles, mainly organic or mixed organic-inorganic? Even those cells located at different depths within the particles may have ended there during filtration if particles are of soft organic consistency. Can the authors exclude the presence of such artifact during filtration?

Line 287: Can the author explain how topography of larger particles (larger than 20 μm) may have impacted the nanoSIMS measurements i.e. the secondary ion yield as nanoSIMS is surface sensitive and requires generally flat surfaces for analysis?

Reviewer #2 (Remarks to the Author):

Harding et al. employed stable isotope incubations followed by nanoscale secondary-ion mass spectrometry to determine rates/activity of non-cyanobacterial diazotrophs (NCD) in the oligotrophic open ocean. As most NCD are not known by their identity, and hence cannot be targeted by standard methods (e.g. CARD-FISH), the authors used an untargeted approach by mapping many random fields of view (to measure random cells) and many particles (to measure particle-associated cells) to statistically encounter NCDs which likely belong to the rare biosphere. The distinction of NCD from cyanobacterial diazotrophs was justified by a dual-isotope approach ($^{15}\text{N}_2$ and $^{13}\text{C-DIC}$) where NCD would have a ^{15}N enrichment only, but no ^{13}C -enrichment, which otherwise would indicate a primary producer. In contrast, cyanobacterial diazotrophs are typically characterized by enrichments in both isotopes. The study was accompanied by molecular analyses of a marker gene for N_2 fixation, *nifH*, which indicated the relative abundance of NCD in a given sample. The authors found several interesting aspects, mainly that NCDs are (almost) exclusively associated with particles and that their abundance using this approach is similar to that found with commonly used quantitative PCR approaches. Further, the extrapolated contribution of NCD to the volumetric rates of N_2 fixation were substantial during this study, where bulk rates were found to be relatively low. In my opinion, the conclusions are solid and the findings substantiate previous hypotheses of particle-associated NCDs. This study is the first (to my knowledge) that shows active N_2 fixation by NCDs, as the authors also point out, and underpins that we have yet to learn a lot about non-cyanobacterial diazotrophs.

My main concerns are related to relatively sparse description of some methods and data presentation. Below I have outlined my comments in more detail.

Specific comments:

I 16: The authors mention that they have measured tens of thousands of cells, which I totally believe given the chances of finding NCDs. However, I think the authors should show those data, which also visualizes just how scarce NCDs are, and yet they contribute substantially to N_2 fixation. This aspect completely vanishes if not shown in a figure. This comment also applies to some sections later in the manuscript where the authors mention this data but it isn't shown anywhere.

I 44/45: None of these citations show the activity by UCYN-A, although by now several articles have shown their activity using nanoSIMS (e.g. Thompson et al. 2012).

I 56-58: There are also some references that actually showed NifH proteins in particles using immunolabeling. I think it is worthwhile to mention these here in the introduction (e.g. Geisler et al. 2019, Sci Rep) even though the type of particles and the type of diazotrophs may or may not be very different.

I 154 (and others later): Please describe the 'automated analysis routine ('chained analysis') in detail.

I 193: If there was no difference in the number of NCD on the particle surface relative to 'inside' the particles, how would this lead to an underestimation using the surface numbers? I guess maybe the authors meant that there was a difference rather than no difference. The difference is described later on (I 304/304).

Figures: Please add some (example) microscopic images (after the DAPI staining) for correlation to the nanoSIMS images, showing individual cells and/or how the measurements of cells were distinguished.

Figure 1: In my opinion, box plots of the N₂ fixation rate would be more informative here but I will leave the decision to the authors.

I 241/242: Possibly this sentence is incomplete?

I 269: An example for comment above, please show the data for these particles, even when no NCD were detected. These don't have to be every individual nanoSIMS image of course, but at least the single-cell data that went into this manuscript.

Figure 2A: It would be great if the images wouldn't show just the at% excess but the actual natural abundance values as well. Also, the legend says that there are four quadrants but there only two halves visible. Please show the data. Further, what kind of particle is this, a cluster of cells, and aggregate of different organisms, detrital material? Please show some microscopic images to correlate with the nanoSIMS images.

[Editor's note: Please find additional comment by this Reviewer regarding Reviewer #3's comments]

After reading the comments by reviewer 3 and re-reading some parts of the manuscript, I reckon that some of the technical criticisms by reviewer 3 are valid while I don't agree with some of them. I will briefly outline my opinion to some of the technical issues raised by reviewer 3 (with NCD referring to non-cyanobacterial diazotrophs or non-cyanobacterial N₂ fixation):

The reviewer has argued that NCD cannot unambiguously be detected due to several aspects:

a. Cellular enrichments of ¹³C could also be the result of biochemical processes other than canonical photosynthesis. While this is true, the authors actually do not consider cells with a significant ¹³C enrichment as NCD but as cyanobacteria (although these could also be other cells). Hence, this approach may lead to false negatives; for example, ¹⁵N-enriched NCD that carried out dark carbon fixation or anaplerotic reactions may be falsely excluded as cyanobacteria. However, false positives are (nearly) completely excluded; for example, a cyanobacterium that only fixed N₂ but no CO₂ during the incubation may be considered an NCD under the authors' analyses; however, the chances of this happening are quite low (maybe the authors could add this aspect to their discussion?). As such this approach is conservative and/or may underestimate NCD. One aspect that the reviewer is correct though, is that the detection limit for a significant ¹³C-enrichment is quite high, therefore low enrichments of ¹³C due to, for example, slower growth rates may falsely be regarded as NCD. This could, however, be addressed by the authors by applying a more conservative detection limit (<< 10% C_{net}) although most NCD cells are already quite close to zero.

b. The reviewer is correct in stating that potential cross-feeding of ¹⁵N- and/or ¹³C-

labeled compounds cannot be excluded, it rarely can. However, ^{15}N -labelled compounds, for example ammonium (such as from *Trichodesmium* which is known to excrete freshly fixed material), would be available to the broader community, and the subsequent uptake of the labelled ammonium would lead to a much larger number (or nearly all) of the cells being ^{15}N -labelled. In contrast, individual, high enrichments of ^{15}N in some cells is a clear signal of N_2 fixation rather than an individual cell thriving on ^{15}N -labelled ammonium that is also available to all other members of the microbial community. Direct transfer of material from cyanobacteria to associated NCD within particles could be excluded by applying some quality-control steps in the analyses. In my opinion, these issue could be resolved through careful analyses of the already existing data (possibly applying an even higher threshold for significant enrichments) and the presentation of the data in the manuscript, which concurs with my request for the missing data and microscopic images to be shown.

c. I do not agree with the statement that NCD cannot be sampled by the authors' approach because it excludes particles >210 micrometer (and the possibility that these particles do not experience any anoxia). The premise that NCD can only occur under anoxia has so far not been shown, it has been modeled and hypothesized. In general, N_2 -fixing microorganisms, including cyanobacteria and heterotrophs, employ various mechanisms to protect their nitrogenase enzyme, for example, the formation of alginate/capsule or respiratory protection. Neither one of these has been shown or refuted for marine NCD. Until then, the assumption that NCD in particles need anoxia for N_2 fixation remains purely speculative and cannot be used as an argument here, in my opinion.

In summary, I think the reviewer raised some valid points, and in retrospect, I could have been more firm in my review regarding the aspect that the missing data and the lack of microscopic images make it difficult to judge some of the authors' claims or, in other words, if the data were shown, the judgement may be different. Assuming that the authors show the missing data etc. in a revised manuscript, and the conclusions/claims are upheld, I still consider this manuscript a valuable addition to the knowledge on marine N_2 fixation.

Reviewer #3 (Remarks to the Author):

Review NCOMMS-21-19855

Harding et al. present a nanoSIMS approach to measure non-cyanobacterial diazotroph (NCD) N_2 fixation rates associated to suspended particles. For more than a decade, several studies, indirect approaches, and reviews have discussed the potential role of particles as microenvironments providing labile organic matter and low oxygen levels suitable for NCD activity in the oligotrophic and oxygenated water column. Yet, no study has succeeded at demonstrating active NCD N_2 fixation in the ocean.

The authors incubate suspended particles with $^{15}\text{N}_2$ and ^{13}C and consider that cells labeled with both isotopes are autotrophic diazotrophs (i.e. cyanobacteria), while cells only labeled with $^{15}\text{N}_2$ are NCDs (non-photosynthetic prokaryotes). While the community has been waiting for NCD N_2 fixation to be demonstrated in natural environments, I regret to say this approach is oversimplistic and cannot unambiguously identify NCD N_2 fixation.

Specific comments

In general, a lack of ancillary data is lacking. DOC, POC, nitrate, oxygen... data should have been provided an used to interpret data accordingly.

Lines 71-74: Why is (Martínez-Pérez et al., 2018) not cited?

Lines 86-91: This approach cannot unambiguously detect NCD N₂ fixation. ¹³C uptake is not only driven by photoautotrophs, chemolithoautotrophic nitrification and anaplerotic pathways may also contribute to bicarbonate fixation in surface waters (Baltar and Herndl, 2019; González et al., 2008; Palovaara et al., 2014; Yool et al., 2007). Dark inorganic carbon fixation contributes up to 36% of primary production in photic waters of stations ALOHA and BATS (Baltar and Herndl, 2019). As for anaplerotic metabolisms, in DOM-poor environments (such as the region studied here, although no DOM data is provided) Bacteroidetes uses inorganic carbon to fulfill its TCA cycle requirements through anaplerotic pathways (González et al., 2008). Importantly, Bacteroidetes show a predominant particle-attached lifestyle. A fast search on OGA for anaplerotic pathway genes shows a wide distribution in surface waters across the oceans, including the region studied here. ¹⁵N enrichment cannot unambiguously be interpreted as NCD N₂ fixation, as cross-feeding may have taken place during the incubation period (i.e. ¹⁵N-labeled NH₄ and DON transferred from co-habitant cyanobacterial diazotrophs -which were also present in the samples- taken up by bacteria).

The potential for NCD N₂ fixation on <210 μm particles collected in a fully oxygenated surface waters is also doubtful. NCDs need an anoxic environment to fix N₂, which is only reached in >0.5 mm particles under such oxic conditions (Bianchi et al., 2018). With nitrate suppressing N₂ fixation, sulfate is more likely used as an electron source for NCD N₂ fixation in anoxic particles (Chakraborty et al., 2020). Sulfate-reduction is predicted to occur in particles >0.5mm, well above the size range sampled in this study (Bianchi et al., 2018). However, ambient nitrate concentrations are not provided.

Why are carbon fixation rates not shown?

Lines 96 and below: The authors must note that Niskin bottles are recognized to underestimate particle concentrations, and result in a bias towards non-sinking particles. NCD N₂ fixation likely occurs in larger particles susceptible to the creation of anoxic microenvironments, which cannot be sampled by this approach. This fact is only worsened by the 210 μm Nitex screening.

Line 166: The natural ¹⁵N enrichment of cells varies largely and non-¹⁵N₂ incubated samples should have been used as a reference instead of Pseudomonas.

Line 178-179: I don't understand what the Pseudomonas were used then for (?).

Lines 184 and below: This extrapolation exercise is prone to several errors due to the particle sampling bias introduced by the sampling approach. Furthermore, the 0.07 mL samples analyzed by the nanoSIMS (Line 255) is certainly not representative enough to try making bigger picture extrapolations.

Lines 236-237: It is puzzling how the authors criticize previous reports of potential NCD N₂ fixation based on nifH amplicon sequence in the introduction, while here they do exactly the same to justify that NCDs contributed significantly to the overall diazotrophic community.

Lines 262-264: I think it mostly illustrates differences in the sampling approaches.

Lines 321-322: and reinforces the possibility of cross-feeding.

References

Baltar, F. and Herndl, G. J.: Ideas and perspectives: Is dark carbon fixation relevant for oceanic primary production estimates?, *Biogeosciences*, 16(19), 3793–3799, 2019.
Bianchi, D., Weber, T. S., Kiko, R. and Deutsch, C.: Global niche of marine anaerobic

metabolisms expanded by particle microenvironments, *Nature Geoscience*, doi:10.1038/s41561-018-0081-0, 2018.

Chakraborty, S., Andersen, K. H., Visser, A. W., Inomura, K., Follows, M. J. and Riemann, L.: Quantifying nitrogen fixation by heterotrophic bacteria in sinking marine particles, *Research Square*, 2–4, 2020.

González, J. M., Fernández-Gómez, B., Fernández-Guerra, A., Gómez-Consarnau, L., Sánchez, O., Coll-Lladó, M., Del Campo, J., Escudero, L., Rodríguez-Martínez, R., Alonso-Sáez, L., Latasa, M., Paulsen, I., Nedashkovskaya, O., Lekunberri, I., Pinhassi, J. and Pedrós-Alió, C.: Genome analysis of the proteorhodopsin-containing marine bacterium *Polaribacter* sp. MED152 (Flavobacteria), *Proc. Natl. Acad. Sci. U. S. A.*, 105(25), 8724–8729, 2008.

Martínez-Pérez, C., Mohr, W., Schwedt, A., Dürschlag, J., Callbeck, C. M., Schunck, H., Dekezemacker, J., Buckner, C. R. T., Lavik, G., Fuchs, B. M. and Kuypers, M. M. M.: Metabolic versatility of a novel N₂-fixing Alphaproteobacterium isolated from a marine oxygen minimum zone, *Environ. Microbiol.*, 20(2), 755–768, 2018.

Palovaara, J., Akram, N., Baltar, F., Bunse, C., Forsberg, J., Pedros-Alio, C., Gonzalez, J. M. and Pinhassi, J.: Stimulation of growth by proteorhodopsin phototrophy involves regulation of central metabolic pathways in marine planktonic bacteria, *Proceedings of the National Academy of Sciences*, 111(35), E3650–E3658, 2014.

Yool, A., Martin, A. P., Fernández, C. and Clark, D. R.: The significance of nitrification for oceanic new production, *Nature*, 447(7147), 999–1002, 2007.

Comments from Reviewer #1:

My main concerns are that (1) evidence that the NCDs are actually associated/attached to particles is not obvious, and (2) I find it unconvincing that ^{15}N enriched hot spots identified and measured under the nanoSIMS are NCDs cells or microbial cells at all.

I believe the study is extremely valuable but is currently incomplete. As much as I understand the difficulties associated with such single-cell analyses, a simple microscopy picture of fluorescence in situ hybridization on one single particle using 16S or 23S rRNA specific nucleotide probe for NCD group to show its association with the particles will provide the direct proof and identification of this group. Such approach, even without being correlated with nanoSIMS analysis of the same particle (which is very difficult to achieve) will still provide a valuable evidence of the presence and association of NCDs with the targeted particles. Without such direct proof (using the suggestion above, or other means to prove that the hot spots indeed represent cells), measurements of ^{15}N hot spots alone cannot be used to present single-cell measurements and build up discussions around single-cell NCDs rates and their importance in the oceanic N-cycling.

Specific comments:

Comment 1; Lines 163-165: main concern regarding cell identification, $^{12}\text{C}^{14}\text{N}$ - secondary ion map is not enough to identify microbial cells inhabiting an organic rich particle containing also $^{12}\text{C}^{14}\text{N}$ molecular ions. Was there a ^{31}P map recorded as proxy for cell nucleoid?

Response: We did not collect ^{31}P during the nanoSIMS analysis, and in our experience, the presence of ^{31}P or ^{32}S , in addition to C and N, cannot be used to determine if an analyzed area is a cell (we find P and S in dead organic material as well as non-organic minerals). In order to address the reviewers concern on NCD being cells, we clarified the text to explain that in addition to $^{12}\text{C}^{14}\text{N}$ -, NCD cells also needed a minimum of $^{15}\text{N}^{12}\text{C}$ - and $^{13}\text{C}^{12}\text{C}$ - counts. The new text states (line 168) “Cells were identified as regions of interest (ROIs) based on a threshold of 150 counts of $^{12}\text{C}^{14}\text{N}$ - per pixel. The data for these ROIs were extracted and then quality controlled by removing data for ROIs with less than 100 total counts for at least one of the minor isotopes or error greater than 30% of the ratio value.” Additionally, to be sure areas with only ^{15}N enrichment were truly cells we reevaluated the data with stricter qualitative threshold which includes; cells to have a clear cell outline, regular cellular shape and enrichment that encompassed the entire cellular area. Description of the new threshold in the text (line 177) “Additionally, we only counted ROIs enriched in ^{15}N if they had clear cellular outlines in $^{12}\text{C}_2$ -, $^{12}\text{C}^{13}\text{C}$ -, $^{12}\text{C}^{14}\text{N}$ - or $^{12}\text{C}^{15}\text{N}$ - and the enrichments were distributed throughout the entire ROI.” The new qualitative threshold filtered out approximately half the previously included NCD cells. Without additional information to determine whether ^{15}N enriched areas were cells, a strict threshold for NCDs that only included ^{15}N enriched ROI with an obvious cell shape was the best way to define cells.

Comment 2; Line 252; 267: can be that due to large volume filtration on relative small surface of the filter that only NCDs that could stick to the particles would stay on the filters and the other were pushed to the filters pores or removed during processing, e.g. by flush freezing. What is the average cell size of a NCDs cell?

Response: We agree with this possibility. To point out that diazotrophs may have ended up on particles randomly through filtration we added the following text (line 318) “Conversely, it is possible diazotrophic cells appear associated with particles when they were actually free-living by randomly landing on particles or being forced into soft particles during filtration. Further studies are needed to validate the presence of NCDs on suspended marine particles. Regardless of whether particle associated or not, the data show NCDs are capable of N_2 fixation in the oxygenated surface ocean.” Additionally, SEM analysis carried out after the review, revealed the silver filter used had more surface roughness than anticipated (not as flat as polycarbonate filters) and was not the most appropriate for analysis of unattached cells which could potentially get buried deeper into the filter material and not reached by the NanoSIMS ion

beam. Therefore, all unattached cell discussion was cut from the manuscript (revised paragraph on line 260) including abundance estimates and our conclusion that the majority of NCDs are particle associated. The manuscript describes the method's inaccuracy for single cells (line 266) "Although we did find other unattached cells, including putative cyanobacteria, we do not make quantitative conclusions about the abundance of free-living NCDs because the 0.2 μm silver filters used for the analysis have micron-scale pores that may have obscured smaller unattached cells in a biased way (Fig. 2)." The manuscript now focuses on NCD on particles without making any conclusion as to the presence and activity of unattached NCDs.

Lastly the average size of the NCD cells was added to the text as (line 290) 'with an average size of $0.8 \pm 0.3 \mu\text{m}$ '.

Comment 3; Line 276 and line 333:

A.) Where is the proof of NCDs cell identification?

Response: We believe that our careful analyses using multiple independent methods (*nifH* sequencing, bulk N_2 fixation, and NanoSIMS particle analyses with dual isotope labeling) provide solid evidence that NCDs fixed N_2 in our incubations, but indeed, at this point we cannot show the phylogenetic identify of the particle-attached N_2 fixing cells. Most marine NCD are only known by the *nifH* gene, making identification via CARD-FISH (a well-established protocol for single cell rate measurements in cyanobacterial diazotrophs which relies on the 16S rRNA gene) currently not achievable. Furthermore, CARD-FISH is group specific, while the goal of this project was to look for all NCD regardless of phylogenetic affiliation. Lastly, although recent studies have targeted the nitrogenase enzyme, and excluded cyanobacterial diazotrophs by the presence of phycoerythrin (Geisler et al., 2019, 2020), staining and non-labeled substance binding (such as antibodies) can drastically reduce the signal of ^{15}N incorporation (Meyer et al., 2021). This could make the already low rates of N_2 fixation by NCDs undetectable. In this study we relied solely on the incorporation of ^{15}N and not ^{13}C to identify NCDs and based on the reviewer's comments have since added the stipulation that there must be a clear cell-like outline in the area of ^{15}N enrichment and the entire cell should show enrichment.

B.) Can the author rule out the ^{15}N enrichment in the cell was not transferred from a cyanobacterial diazotroph within the 24h time of isotope labeling?

Response: Thank you for pointing this out and we agree this uncertainty should be addressed in the manuscript. Therefore, we included text supporting why secondary transfer is unlikely to be measured as a false positive in our study. New text (line 341) "While it is possible the new N biomass synthesized by NCDs may be due to secondary uptake of ^{15}N labeled NH_4 derived from cyanobacteria, it is unlikely those values would be high enough to pass the conservative minimum ^{15}N threshold used to define an N_2 -fixing cell given the initial amount of ^{15}N added to the incubation ($2.8 \text{ At}\% \pm 0.4$). Additionally, the few high ^{15}N enrichment values measured in this study are highly improbable to have resulted from secondary uptake during the 24-hour

incubation.” Secondary transfer is always a concern in isotope studies and can rarely be ruled out, yet we believe the conservative threshold we set and low initial ^{15}N label available to the cells should minimize the possibility of this occurrence. Secondly, we measured several high values of NCD N_2 -fixation that would be nearly impossible to achieve by secondary transfer during the length of the incubation.

C.) Can the authors prove that the ^{15}N hot spots in the particle are microbial cells at all?

Response: It is unlikely that ^{15}N hotspots could be attributed to anything other than a microbial cell. The $^{15}\text{N}_2$ gas is initially added to the incubations and any remaining $^{15}\text{N}_2$ gas that had not been incorporated into cellular biomass after 24 hours is removed by filtration which only collects particulate matter ($>0.2\ \mu\text{m}$). Unlikely, but if any $^{15}\text{N}_2$ gas was not removed during filtration it would not be measured by nanoSIMS as the nanoSIMS instrument collects $^{12}\text{C}^{15}\text{N}$. As the ^{15}N must be bound to a C it is highly likely that the ^{15}N had been incorporated into cellular biomass. Additionally, as evidence of cell presence on particles we have included new figures (Fig. 2 and Fig. S2) that show particles with attached cells via SEM, $^{12}\text{C}_2^-$ and DAPI staining.

Comment 4; Line 282:

A) “The majority of NCDs (66 cells) were clearly particle-attached..”

Besides the identification issue pointed out above, as the water samples were deposited on the filter by filtration of a large volume of 1 L or higher how one can claim that cells on the particle surface are particle attached and not accidentally attached during filtration? What is the composition of the particles, mainly organic or mixed organic-inorganic?

Even those cells located at different depths within the particles may have ended there during filtration if particles are of soft organic consistency. Can the authors exclude the presence of such artifact during filtration?

Response: We appreciate this point by the reviewer, and we agree that we cannot be completely certain the cells did not randomly land on the particles during filtration. We have searched the literature for any publication on this topic (for example, fixing axenic protists and bacteria separately, then mixing them together and filtering them to see if there is artefactual attachment; sadly we cannot find such a study). Thus, to be conservative, we highlight this possibility by adding the following statement to the text (line 318) **“Conversely, it is possible diazotrophic cells appear associated with particles when they were actually free-living by randomly landing on particles or being forced into soft particles during filtration. Further studies are needed to validate the presence of NCDs on suspended marine particles. Regardless of whether particle associated or not, the data show NCDs are capable of N_2 fixation in the oxygenated surface ocean.”** As mentioned in response to comment 2 by this reviewer we have edited the text to remove all discussion and calculations regarding unattached cells (revised paragraph on line 260). The method used in this study is biased towards cells on particles and

therefore we cannot make any statements regarding unattached cells, or the possibility that NCDs are strictly particle associated. With this modification, whether the cells are associated with particles or merely landed on a particle during filtration becomes less of a concern as the main conclusion is that NCD fix N_2 in the surface ocean.

Although we do not know the specific composition of the particles, we added both $^{12}C_2^-$ and $^{12}C^{14}N^-$ images from a representative particle to show the organic nature of the particles analyzed (Fig. 2c and Fig. 2d). Additionally, we added text stating the unknown nature of the particles (line 323). “Particles available for analysis ranged from 5 to 210 μm in diameter (pore size of the incubator pre-filter) and included densely packed particles (Fig. 2, Fig. S2) as well as loosely formed aggregates (Fig. S2). The particle compositions and internal nutrient concentrations are unknown although high concentration of both $^{12}C_2^-$ and $^{12}C^{14}N^-$ measured by nanoSIMS show the particles have a high organic content (Fig. 2c and d).”

Comment 5; Line 287: Can the author explain how topography of larger particles (larger than 20 μm) may have impacted the nanoSIMS measurements i.e. the secondary ion yield as nanoSIMS is surface sensitive and requires generally flat surfaces for analysis?

Thank you for the question, topography was not be an issue with our samples as the particles chosen for analysis were relatively small, organic flocs, and were flattened during filtration, ensuring the maximum topography of the analyzed particles (<10 μm , typically less than 2 μm) is below the acceptable value for nanoSIMS. Topography should be below 20 μm (possible 30 μm but with loss of precision) for isotopic enrichments experiments analyzed with nanoSIMS such as the one used here (Mueller et al., 2013). A selection of particles checked with SEM show topography was not more than 10 microns, and their aspect ratios (height to width) were low. In our experimental design, we filtered out larger particles using a prefilter of 210 μm . Additionally, particles were only chosen for nanoSIMS analysis that fit within the real-time imaging window (using secondary electrons) of 150 by 150 μm . Furthermore, the average size of particles that contained at least 1 NCD was $14 \pm 7 \mu m$ (average excludes 1 particle >20 μm). At each particle location, we did automatic peak centering both for secondary beam alignment and mass to ensure correct turning for each particle individually. Additionally, aside from the 9 particles measured with an associated NCD we have ~961 particles (150-5 μm) that do not show ^{15}N enrichment akin to an NCD cell.

Comments from Reviewer #2:

My main concerns are related to relatively sparse description of some methods and data presentation. Below I have outlined my comments in more detail.

Comment 1; l 16: The authors mention that they have measured tens of thousands of cells,

which I totally believe given the chances of finding NCDs. However, I think the authors should show those data, which also visualizes just how scarce NCDs are, and yet they contribute substantially to N₂ fixation. This aspect completely vanishes if not shown in a figure. This comment also applies to some sections later in the manuscript where the authors mention this data but it isn't shown anywhere.

Response: Thank you for pointing out the importance of including all the data. We added all the data (n > 41,000 cells) into the plot of Fig. 3b.

Comment 2; 144/45: *None of these citations show the activity by UCYN-A, although by now several articles have shown their activity using nanoSIMS (e.g. Thompson et al. 2012).*

Response: Thank you for bringing this to our attention. We have now added the appropriate references for cyanobacterial single cell rates. The new correct references are: (line 45) “single cell analyses (Foster et al., 2011; Gradoville et al., 2020; Martínez-Pérez et al., 2016; Thompson et al., 2012).”

Comment 3; 156-58: *There are also some references that actually showed NifH proteins in particles using immunolabeling. I think it is worthwhile to mention these here in the introduction (e.g. Geisler et al. 2019, Sci Rep) even though the type of particles and the type of diazotrophs may or may not be very different.*

Response: We agree and have modified line 58 to read: Heterotrophic N₂ fixation has been suggested to occur in or on particles (Bombar et al. 2013; Matthew J Church et al. 2005; H. W. Paerl and Prufert 1987; Pedersen et al. 2018; Lasse Riemann, Farnelid, and Steward 2010, Geisler et al. 2019). Note that we did include Geisler in the introduction at a later point as a relevant reference for the mentioned hypothesis of NCD being attached to particles.

Comment 4; 1154 (and others later): *Please describe the ‘automated analysis routine (“chained analysis”) in detail.*

Response: We have revised the text to include a more thorough description of the “chained analysis” which we have simplified to automated analysis routine. The new description states, (line 159) “ Each analysis took 35 to 65 minutes of instrument time. NanoSIMS measurements were conducted with both untargeted and particle-targeted approaches. The untargeted approach used an automated analysis routine which sequentially moved the stage to scan non-overlapping raster areas to map contiguous areas of the filter, and at each new location an automated peak centering was redone before data collection. To target particles, we first mapped a large number of them using nanoSIMS secondary electron imaging and then we used the same automated analysis routine to target particle locations.”

Comment 5; 1193: *If there was no difference in the number of NCD on the particle surface relative to ‘inside’ the particles, how would this lead to an underestimation using the surface*

numbers? I guess maybe the authors meant that there was a difference rather than no difference. The difference is described later on (l 304/304).

Response: Thank you for pointing out this confusing wording. After consideration we decided using two methods for one calculation is overly confusing and have simplified the method and discussion section by only using one calculation. Therefore, this sentence was removed as well as all other mentions of the particle volume method to calculate NCD bulk contributions.

Comment 6; Figures: *Please add some (example) microscopic images (after the DAPI staining) for correlation to the nanoSIMS images, showing individual cells and/or how the measurements of cells were distinguished.*

Response: We agree and have added SEM, $^{12}\text{C}_2^-$ and $^{12}\text{C}^{14}\text{N}^-$ images to the main text in Fig. 2a-e. We have additionally added DAPI-stained particle images to the supplementary text (Fig. S2a-d), showing examples of the types of particles analyzed as well as attached cells.

Comment 7; Figure 1: *In my opinion, box plots of the N_2 fixation rate would be more informative here but I will leave the decision to the authors.*

Response: Thank you for the suggestion but we feel the overlapping bar plot is an easily interpretable visualization of the data comparing natural-light to all-dark N_2 -fixation rates by station (Fig. 1).

Comment 8; l 241/242: *Possibly this sentence is incomplete?*

Response: Thank you for pointing this out, line 247 now reads: “**The all-dark N_2 fixation** rates at stations 17 and 20 were below MQR, while the all-dark data for station 23 was not available.”

Comment 9; l 269: *An example for comment above, please show the data for these particles, even when no NCD were detected. These don't have to be every individual nanoSIMS image of course, but at least the single-cell data that went into this manuscript.*

Response: Thank you, we agree including all data shows a more comprehensive picture of the study. We added all the data, enriched and unenriched cells ($n > 41,000$) into the plot in Fig. 3b.

Comment 10; Figure 2A: *It would be great if the images wouldn't show just the at% excess but the actual natural abundance values as well. Also, the legend says that there are four quadrants but there only two halves visible. Please show the data. Further, what kind of particle is this, a cluster of cells, and aggregate of different organisms, detrital material? Please show some microscopic images to correlate with the nanoSIMS images.*

Response: We have incorporated all these suggestions. We have added grey arrows to the color

scale bars of the nanoSIMS images to indicate natural abundance levels in Fig. 2a and Fig. 3a. The caption of Fig. 3b is now amended to read 'Color scale bars correspond to At% enrichment for the given isotope, *grey arrow indicate natural abundance of rare isotope.*' Additionally, incorporating all the data into Fig. 3b has made all four quadrants easily visible. Lastly, as mentioned, we have included particle examples images by SEM (Fig. 2b&e) and DAPI-stained particle examples with attached cells in supplementary material Fig. S2.

[Editor's note: Please find additional comment by this Reviewer regarding Reviewer #3's comments]

After reading the comments by reviewer 3 and re-reading some parts of the manuscript, I reckon that some of the technical criticisms by reviewer 3 are valid while I don't agree with some of them. I will briefly outline my opinion to some of the technical issues raised by reviewer 3 (with NCD referring to non-cyanobacterial diazotrophs or non-cyanobacterial N₂ fixation):

The reviewer has argued that NCD cannot unambiguously be detected due to several aspects:

a. Cellular enrichments of ¹³C could also be the result of biochemical processes other than canonical photosynthesis. While this is true, the authors actually do not consider cells with a significant ¹³C enrichment as NCD but as cyanobacteria (although these could also be other cells). Hence, this approach may lead to false negatives; for example, ¹⁵N-enriched NCD that carried out dark carbon fixation or anaplerotic reactions may be falsely excluded as cyanobacteria. However, false positives are (nearly) completely excluded; for example, a cyanobacterium that only fixed N₂ but no CO₂ during the incubation may be considered an NCD under the authors' analyses; however, the chances of this happening are quite low (maybe the authors could add this aspect to their discussion?). As such this approach is conservative and/or may underestimate NCD. One aspect that the reviewer is correct though, is that the detection limit for a significant ¹³C-enrichment is quite high, therefore low enrichments of ¹³C due to, for example, slower growth rates may falsely be regarded as NCD. This could, however, be addressed by the authors by applying a more conservative detection limit (<< 10% Cnet) although most NCD cells are already quite close to zero.

Response: We greatly appreciate the additional reviews here regarding Reviewer #3's comments. We hope our edits satisfy them. Regarding what we call cyanobacteria-like cells being NCDs carrying out dark carbon fixation or anaplerotic C fixation (false negatives), it is unlikely that these processes would be detected by our methods, since the amount of ¹³C DIC added was relatively low, so C fixation needed to be relatively high to be detected. However, we agree with the reviewer that these cells could be non-cyanobacteria, but still autotrophs, and in this study would be classified as cyanobacterial-like cells, leading to an underestimate of NCD cells and N₂ fixation rates. In that respect, our approach was designed to have confidence that the ¹⁵N enriched cell that lacked ¹³C enrichment were NCDs, although possibly leading to

an underestimation, our approach does not assume to measure any and all NCDs present.

*b. The reviewer is correct in stating that potential cross-feeding of ^{15}N - and/or ^{13}C -labeled compounds cannot be excluded, it rarely can. However, ^{15}N -labelled compounds, for example ammonium (such as from *Trichodesmium* which is known to excrete freshly fixed material), would be available to the broader community, and the subsequent uptake of the labelled ammonium would lead to a much larger number (or nearly all) of the cells being ^{15}N -labelled. In contrast, individual, high enrichments of ^{15}N in some cells is a clear signal of N_2 fixation rather than an individual cell thriving on ^{15}N -labelled ammonium that is also available to all other members of the microbial community. Direct transfer of material from cyanobacteria to associated NCD within particles could be excluded by applying some quality-control steps in the analyses. In my opinion, these issue could be resolved through careful analyses of the already existing data (possibly applying an even higher threshold for significant enrichments) and the presentation of the data in the manuscript, which concurs with my request for the missing data and microscopic images to be shown.*

Response: We agree, and again, we hope that the more conservative qualitative thresholds and the microscopic images will resolve these concerns.

c. I do not agree with the statement that NCD cannot be sampled by the authors' approach because it excludes particles >210 micrometer (and the possibility that these particles do not experience any anoxia). The premise that NCD can only occur under anoxia has so far not been shown, it has been modeled and hypothesized. In general, N_2 -fixing microorganisms, including cyanobacteria and heterotrophs, employ various mechanisms to protect their nitrogenase enzyme, for example, the formation of alginate/capsule or respiratory protection. Neither one of these has been shown or refuted for marine NCD. Until then, the assumption that NCD in particles need anoxia for N_2 fixation remains purely speculative and cannot be used as an argument here, in my opinion.

Response: We appreciate this insight, and we agree.

In summary, I think the reviewer raised some valid points, and in retrospect, I could have been more firm in my review regarding the aspect that the missing data and the lack of microscopic images make it difficult to judge some of the authors' claims or, in other words, if the data were shown, the judgement may be different. Assuming that the authors show the missing data etc. in a revised manuscript, and the conclusions/claims are upheld, I still consider this manuscript a valuable addition to the knowledge on marine N_2 fixation.

Comments from Reviewer #3:

Review NCOMMS-21-19855

Harding et al. present a nanoSIMS approach to measure non-cyanobacterial diazotroph (NCD) N₂ fixation rates associated to suspended particles. For more than a decade, several studies, indirect approaches, and reviews have discussed the potential role of particles as microenvironments providing labile organic matter and low oxygen levels suitable for NCD activity in the oligotrophic and oxygenated water column. Yet, no study has succeeded at demonstrating active NCD N₂ fixation in the ocean.

The authors incubate suspended particles with ¹⁵N₂ and ¹³C and consider that cells labeled with both isotopes are autotrophic diazotrophs (i.e. cyanobacteria), while cells only labeled with ¹⁵N₂ are NCDs (non-photosynthetic prokaryotes). While the community has been waiting for NCD N₂ fixation to be demonstrated in natural environments, I regret to say this approach is oversimplistic and cannot unambiguously identify NCD N₂ fixation.

Specific comments

In general, a lack of ancillary data is lacking. DOC, POC, nitrate, oxygen... data should have been provided and used to interpret data accordingly.

Response: We agree and have included nutrient and oxygen concentration information to the text, (line 105) “Nutrient concentrations (NO₃⁻ + NO₂ and PO₄³⁻) were analyzed according to EPA protocol (40 CFR part 136, appendix B), and the method limit of detections were LOD; 0.01 μM and 0.02 μM, respectively.” (line 293) “All samples, including those containing active NCD cells, were from fully oxygenated surface waters” and (line 300) “The presence of NCDs showed no relationship to bulk seawater nutrient concentrations (nitrate: 0.1 μM, 0.1 μM, <LOD, <LOD, 0.2 μM, <LOD, <LOD, ordered by station; phosphate <LOD at all stations.”

Comment 1; Lines 71-74: Why is (Martínez-Pérez et al., 2018) not cited?

Response: Thank you for pointing out this oversight. This reference is now included in the text (line 71) “Marine NCD N₂ fixation rate measurements are limited to a few cultured representatives from estuarine environments with rates of 0.02 to 1.1 fmol N cell⁻¹ d⁻¹ (scaled to per day rates assuming 24 hours of N₂ fixation) (Bentzon-Tilia et al., 2015b; Martinez-Perez et al., 2017; Paerl et al., 2018).”

Comment 2;

A.) Lines 86-91: This approach cannot unambiguously detect NCD N₂ fixation. ¹³C uptake is not only driven by photoautotrophs, chemolithoautotrophic nitrification and anaplerotic pathways may also contribute to bicarbonate fixation in surface waters (Baltar and Herndl, 2019; González et al., 2008; Palovaara et al., 2014; Yool et al., 2007). Dark inorganic carbon fixation contributes up to 36% of primary production in photic waters of stations ALOHA and BATS (Baltar and Herndl, 2019). As for anaplerotic metabolisms, in DOM-poor environments (such as

the region studied here, although no DOM data is provided) Bacteroidetes uses inorganic carbon to fulfill its TCA cycle requirements through anaplerotic pathways (González et al., 2008). Importantly, Bacteroidetes show a predominant particle-attached lifestyle. A fast search on OGA for anaplerotic pathway genes shows a wide distribution in surface waters across the oceans, including the region studied here.

Response: Note that this comment is related to false negatives (also discussed above by reviewer #2, comment a), the possibility that what we call “cyanobacterial-like diazotrophs” are actually NCDs that were fixing carbon either autotrophically or anaplerotically. This does not impact our data of what we call NCDs (false positives are discussed below). We agree that these mechanisms by which NCDs could incorporate $^{13}\text{CO}_2$ are possible, but our study only focused on the cells that did not incorporate $^{13}\text{CO}_2$. In that respect, our approach was designed to have confidence that the ^{15}N enriched cell that lacked ^{13}C enrichment were NCDs, although possibly leading to an underestimation, our approach does not assume to measure any and all NCDs present. As the reviewer states, NCDs fixing N_2 and incorporating ^{13}C through chemolithoautotrophy or anaplerotic pathways could likely be falsely classified as cyanobacterial-like diazotrophs, leading to an underestimate of NCD cells and their contribution to N_2 fixation rates. However, we used conservative thresholding criteria to identify cells as NCDs. The NCD ^{13}C carbon “enrichment” ranged from -2.7 to 1.9% of newly fixed ^{13}C with an average of 0.29% (± 1.04), indicating very low ^{13}C enrichment if any. So, although ^{13}C incorporation by NCDs is possible, our study show N_2 fixation without ^{13}C is occurring in the surface ocean. Additionally, any false negatives as described would result in an underestimation of the in-situ counts, which is preferable to an overestimate of NCD abundances.

Cyanobacterial-like diazotrophs that did not fix C and thus did not incorporate ^{13}C over the incubation period (24-hrs) is a possibility and is discussed in the manuscript (line 428) “As such, the lower cyanobacterial abundances may be due to the possibility of typically C-fixing cyanobacteria being able to use heterotrophic C acquisition while attached to particles and would therefore be categorized as a NCD in this study. Both *Trichodesmium* and *Cyanothece* have shown evidence of mixotrophy by incorporating dissolved organic C while maintaining their N_2 fixing capability (Benavides et al., 2017; Feng et al., 2010).”

B.) ^{15}N enrichment cannot unambiguously be interpreted as NCD N_2 fixation, as cross-feeding may have taken place during the incubation period (i.e. ^{15}N -labeled NH_4 and DON transferred from co-habitant cyanobacterial diazotrophs -which were also present in the samples- taken up by bacteria).

Response (same as above to reviewer #1, comment 3B) Thank you for pointing this out and we agree this uncertainty should be addressed in the manuscript. Therefore, we included text supporting why secondary transfer is unlikely to be measured as a false positive in our study. New text (line 341) “While it is possible the new N biomass synthesized by NCDs may be due to secondary uptake of ^{15}N labeled NH_4 derived from cyanobacteria, it is unlikely those values

would be high enough to pass the conservative minimum ^{15}N threshold used to define an N_2 -fixing cell given the initial amount of ^{15}N added to the incubation ($2.8 \text{ At}\% \pm 0.4$). Additionally, the few high ^{15}N enrichment values measured in this study are highly improbable to have resulted from secondary uptake during the 24-hour incubation.” Secondary transfer is always a concern in isotope studies and can rarely be ruled out, yet we believe the conservative threshold we set and low initial ^{15}N label available to the cells should minimize the possibility of this occurrence. Secondly, we measured several high values of NCD N_2 -fixation that would be nearly impossible to achieve by secondary transfer during the length of the incubation.

C.) *The potential for NCD N_2 fixation on <210 μm particles collected in a fully oxygenated surface waters is also doubtful. NCDs need an anoxic environment to fix N_2 , which is only reached in >0.5 mm particles under such oxic conditions (Bianchi et al., 2018). With nitrate suppressing N_2 fixation, sulfate is more likely used as an electron source for NCD N_2 fixation in anoxic particles (Chkraborty et al., 2020). Sulfate-reduction is predicted to occur in particles >0.5mm, well above the size range sampled in this study (Bianchi et al., 2018). However, ambient nitrate concentrations are not provided.*

Response: Regarding the reviewer’s point, it is hypothesized that NCD N_2 fixation is linked to microaerobic zones within particles, but this has not yet been shown as a strict requirement. Although particles larger than 0.6 mm have been hypothesized and modeled to allow for microaerobic zones suitable for anaerobic N_2 fixation, it remains theoretical (Chkraborty et al., 2021; Ploug et al., 2001; Klawonn et al., 2015). Additionally, however likely low-oxygen zones may be to harbor anaerobic N_2 fixation, no data yet exist that exclude the possibility of aerobic N_2 fixation by NCDs such as found in cyanobacterial diazotrophs. In fact, the most commonly occurring NCD group (defined based on the *nifH* gene and known as Gamma A) shows a positive correlation with oxygen concentrations (Langlois et al., 2017). Virtually nothing is known about the physiology of most open ocean NCDs, therefore assuming aerobic N_2 fixation is not possible has no basis at this point in time given the data that are available. However, we also cannot rule out that the particles where we measured NCD N_2 fixation may have been large enough to harbor an anoxic zone but fractured during filtration through the prefilter. We address this final point in (line 412) “If microanoxic zones are required for N_2 fixation, it is possible small particles with associated NCD may have initially been large enough to harbor microaerobic zones but were fractured during filtration or analysis preparation.”

Although we do not know the chemical parameters of the individual particles such as nitrate concentrations, particle biochemistry is dynamic (Chkraborty et al., 2021; Klawonn et al., 2015) and would likely be highly variable over a 24 hour incubation. If nitrate does inhibit N_2 fixation by NCDs (which is not shown) the enrichment of ^{15}N cells shows the particle was at least transiently nitrate limited.

D.) *Why are carbon fixation rates not shown?*

Response: We appreciate the reviewer's suggestion but do not think C fixation rates were relevant to our findings as the paper's focus is on non-C fixing NCDs.

Comment 3; Lines 96 and below: The authors must note that Niskin bottles are recognized to underestimate particle concentrations, and result in a bias towards non-sinking particles. NCD N₂ fixation likely occurs in larger particles susceptible to the creation of anoxic microenvironments, which cannot be sampled by this approach. This fact is only worsened by the 210 µm Nitex screening.

Response: Thank you for pointing this out and we agree this is important information to include in the paper. As such we have added the following additional text (line 393) **Additionally, Niskin bottles are likely to under-sample particles, especially fast-sinking ones (Suter et al., 2017), while use of the 210 µm prefilter further reduces the particle concentrations used to estimate volumetric NCD abundances.**"

Comment 4; Line 166: The natural ¹⁵N enrichment of cells varies largely and non-¹⁵N₂ incubated samples should have been used as a reference instead of Pseudomonas.

Response: We appreciate the reviewer's concern, but we believe use of the *Pseudomonas* reference cells more than accounts for difference in natural abundance of ¹⁵N. The natural abundance of ¹⁵N had an average value of 0.366 ± 0.0005 At%. The equation we used to determine whether a cell is significantly enriched was the average ¹⁵N At% + 3x standard deviation of the average. If we used the environmental natural abundance the enrichment minimum threshold would be 0.368 At%, while the *Pseudomonas* reference cells using the same threshold equation resulted in a minimum threshold of 0.47 At%. We thought the more conservative threshold was appropriate for this study. But in line with the reviewer's comment, we did use the environmental ¹⁵N natural abundance at each station when calculating cell specific N₂ fixation rates once the cells had been defined as enriched based on the *Pseudomonas* threshold to provide a more accurate account of the N₂ fixation rates.

Comment 5; Line 178-179: I don't understand what the Pseudomonas were used then for (?).

Response: Thank you for pointing out this misunderstanding. We have added additional details (in blue) to this description to clarify any confusion, (line 187) "Cell-specific N₂ fixation rates (fmol N cell⁻¹ d⁻¹) were calculated **for cells that were above the minimum enrichment threshold (>0.47 At% ¹⁵N, as described above) following Krupke et al. (2015). The average initial bulk isotopic composition of the unlabeled ("natural abundance") samples from each location were subtracted from the At% ¹⁵N values of enriched cells to estimate rare isotope incorporation...**". In short the *Pseudomonas* cells were used to set a conservative threshold of enriched versus non-enriched cells. If a cell passed the ¹⁵N enrichment threshold set by *Pseudomonas*, we then calculated cell-specific N₂ fixation rates using the station specific ¹⁵N natural abundance which was subtracted from the cell's measured ¹⁵N enrichment value from nanoSIMS.

Comment 6; Lines 184 and below: This extrapolation exercise is prone to several errors due to the particle sampling bias introduced by the sampling approach. Furthermore, the 0.07 mL samples analyzed by the nanoSIMS (Line 255) is certainly not representative enough to try making bigger picture extrapolations.

Response: We agree with reviewer on these points. The extrapolation calculations are surely biased for several reasons, including the particle sampling approach as well as nanoSIMS analysis being restricted to particle surfaces. We think these calculations are informative to include in this study as context, however, we repeatedly call them “rough estimates” and are not presented as absolute values for the water masses sampled. We point out the shortcomings of the extrapolations and emphasize that they are underestimates. The text highlights this point (line 384) “The calculated contributions of NCD activity to total fixed N discussed above are based on analyses of only several thousands of cells from seven stations on one cruise and a number of assumptions, and therefore are only rough estimates of the potential importance of this underexplored phenomenon in the world’s oceans.”

Additionally in line with the reviewer’s comment on unattached cells in the 0.07 ml of water scanned, we have removed any extrapolations concerning unattached cells (revised paragraph on line 260) including abundance estimates and our conclusion that the majority of NCDs are particle associated. The manuscript describes the method’s inaccuracy for single cells and focuses on NCDs on particles without making any conclusion as to the presence of activity of unattached NCDs.

Comment 7; Lines 236-237: It is puzzling how the authors criticize previous reports of potential NCD N₂ fixation based on nifH amplicon sequence in the introduction, while here they do exactly the same to justify that NCDs contributed significantly to the overall diazotrophic community.

Response: We apologize if our discussion of this topic was interpreted as critical, that was not our aim. Our goal was to convey that *nifH* gene sequencing provides evidence of nitrogenase potential but cannot be assumed to be N₂ fixation activity which is why we employed nanoSIMS in addition to *nifH* gene sequencing. To better explain our point less critically we edited the text, (line 49) “The presence of diverse *nifH* genes from NCDs suggests NCD N₂ fixation may be an important process in the euphotic zone, however it has not yet been directly demonstrated that these marine NCDs fix N₂, which is a critical first step for determining their contribution to measured community N₂ fixation rates.”

Comment 8; Lines 262-264: I think it mostly illustrates differences in the sampling approaches.

Response: We agree with the reviewer and have adjusted the text to reflect this (line 281) “Furthermore, although based on a different method to estimate abundances, unattached NCD abundances (< 1.6 or 3 μm) in the oligotrophic North Pacific are reported orders of magnitude

higher when estimated with a primer-free metagenomic based approach...”. We believe this information is important to include as discrepancies between the two quantification methods have not been resolved.

Comment 9; Lines 321-322: and reinforces the possibility of cross-feeding.

Response: We appreciate the reviewers comment and have adjusted the text for clarity as the comparison between the two isotopes is only relevant in regards to cyanobacterial-like cells (line 347) “For the cyanobacterial-like cells, Nnet% and Cnet% were not equivalent, indicating the cells were obtaining either their N or C requirements from other unlabeled sources.” The comparison for ¹⁵N to ¹³C is only relevant for cyanobacteria, because we argue NCDs are not fixing C and as such obtain their C from alternate sources (not ¹³C-bicarbonate). For the cyanobacterial-like diazotrophs, the trend of most cells when comparing percent new ¹⁵N to percent new ¹³C (shown in the graph below) is that cyanobacterial cells have more ¹³C percent biomass than ¹⁵N, which if anything, argues against ¹⁵N cross feeding. A higher percent of new ¹³C than ¹⁵N indicates the cyanobacterial-like cell may be uptaking alternative non-labeled N sources.

References

Chakraborty, S., Andersen, K. H., Visser, A. W., Inomura, K., Follows, M. J. and Riemann, L.: Quantifying nitrogen fixation by heterotrophic bacteria in sinking marine particles, *Nat. Commun.*, 12(1), doi:10.1038/s41467-021-23875-6, 2021.

Geisler, E., Bogler, A., Rahav, E. and Bar-Zeev, E.: Direct detection of heterotrophic diazotrophs associated with planktonic aggregates, *Scientific Reports*, 1–9, doi:10.1038/s41598-019-45505-4, 2019.

Geisler, E., Bogler, A., Bar-zeev, E., Rahav, E. and Farnelid, H.: Heterotrophic Nitrogen Fixation at the Hyper-Eutrophic Qishon River and Estuary System, *Front. Microbiol.*, 11(June), 2012–2021, doi:10.3389/fmicb.2020.01370, 2020.

Klawonn, I., Bonaglia, S., Bruchert, V. and Ploug, H.: Aerobic and anaerobic nitrogen transformation processes in N₂-fixing cyanobacterial aggregates, *ISME J.*, 9(15), 1456–1466, doi:10.1038/ismej.2014.232, 2015.

Langlois, R., Großkopf, T., Mills, M., Takeda, S. and LaRoche, J.: Widespread Distribution and Expression of Gamma A (UMB), an Uncultured, Diazotrophic, γ -Proteobacterial nifH Phylotype, *PLoS One*, 10(6), e0128912, doi:10.1371/journal.pone.0128912, 2015.

Meyer, N. R., Fortney, J. L. and Dekas, A. E.: NanoSIMS sample preparation decreases isotope enrichment: magnitude, variability and implications for single-cell rates of microbial activity, *Environ. Microbiol.*, 23(1), 81–98, doi:10.1111/1462-2920.15264, 2021.

Mueller, C. W., Weber, P. K., Kilburn, M. R., Hoeschen, C., Kleber, M. and Pett-Ridge, J.: Advances in the Analysis of Biogeochemical Interfaces: NanoSIMS to Investigate Soil Microenvironments, *Adv. Agron.*, 121, 1–46, doi:10.1016/B978-0-12-407685-3.00001-3, 2013.

Ploug, H.: Small-scale oxygen fluxes and remineralization in sinking aggregates, *Limnol. Oceanogr.*, 46(7), 1624–1631, doi:10.4319/lo.2001.46.7.1624, 2001.

Suter, E. A., Scranton, M. I., Chow, S., Stinton, D., Faull, L. M. and Taylor, G. T.: Niskin bottle sample collection aliases microbial community composition and biogeochemical interpretation, *Limnol. Oceanogr.*, (62), 606–617, doi:10.1002/lno.10447, 2016.

REVIEWER COMMENTS

Reviewer #1 (Remarks to the Author):

I consider the authors have dedicated huge effort to improve and revise the manuscript. From my point of view they have addressed thoroughly all the raised concerns and criticism. I support the publication of the current revised version.

Reviewer #2 (Remarks to the Author):

After the first round of review, I was quite hopeful that the authors would be able to provide the missing images and, with that, the missing evidence that the 'cellular' measurements are indeed from cells. Unfortunately, I must say that the revised version of the manuscript still falls short in providing the necessary evidence due to a lack of correlative imaging. The correlative imaging approach is the vital link between a cell and its activity, especially when cells are embedded in a matrix like, for example, the particles analyzed here. I do not think that the authors approach to use $^{12}\text{C}^{14}\text{N}$ - ion counts to identify cells (even with the added constraints on the minor isotopes) is a well-founded approach. Even a combination of C,N,S and/or P imaging (as pointed out by reviewer 1) could have helped a little bit here, although some correlative imaging would have still be needed. As such, the only piece of evidence that argues for non-cyanobacterial, or any N_2 fixation on the particles for that matter, is that high enrichments of ^{15}N can only come from N_2 fixation activity in the incubated water. But is this enough for publication? Or should the authors maybe invest the time for some good-quality imaging that provides the evidence that measurements are indeed representative for cells, both for the NCD as well as all other cellular measurements (how can one see that the other ~ 41000 measurements are all cells and not just random biomass?). I think it would be great if it could be shown that particles are indeed sites of N_2 fixation but I also think that the evidence needs to be a bit more convincing at this point (also considering some of the concerns mentioned by reviewers 1 and 3).

Further comments:

I 171: Using a 30% error cut-off is fairly lenient/generous rather than conservative. Would the data and conclusion hold up if the authors used a more conservative cut-off of 5-10% (or maybe even 15%), which is mostly used in nanoSIMS studies? The authors could at least state how many 'cells' actually had errors that high, how many had errors quite far below the 30%?

I 192: The authors determined the C and N content of relatively small cells using the equations provided by Verity et al. 1992. While these conversion factors are good for larger cells, small cells tend to have a higher C density (Khachikyan et al. 2019). Using the higher C content (and with that, also N content), more reflective of the bacterial biomass analyzed here, the cellular rates would be somewhat higher.

Figures 2, 3 and Supp. Fig 3: A threshold for the exclusion of data that goes beyond 'normal' processing seems to have been applied; however, this is not mentioned in the methods and/or figure legends. Regular nanoSIMS imaging does not produce the smooth edges seen here. Please add a disclaimer and/or, preferably, show the less-processed data.

References

Khachikyan, A., Milucka, J., Littmann, S., Ahmerkamp, S., Meador, T., Koenneke, M., Burg, T. and Kuypers, M.M., 2019. Direct cell mass measurements expand the role of small microorganisms in nature. *Applied and Environmental Microbiology*, 85(14), pp.e00493-19.

Reviewer #1 was asked to comment on the report from reviewer #2. These are their remarks:

1. Do you disagree with any of their technical criticisms?

I believe reviewer 2 criticism is fully justified. Presently the authors did not provided direct evidence that the small isotopically enriched spots/particles are actually cells. Even with the reviewer 2 recommendations to use a more conservative cut-off (15%) this situation will not change. It will probably just decrease the number of enriched cell-like particles considered in the analysis. The direct link of function to cell identity can be only solved by combination of FISH with nanoSIMS. However it is almost impossible to apply such correlation to the huge number of samples the authors analyzed (over 41.000 measurements).

The explanation of the authors against FISH identification that may result in a dilution of the isotopic enrichment of the cell-like particles is just. It was shown also previously, that FISH based identification decrease the isotopic enrichment and results in underestimation of the real cellular activity/uptake rate. Nevertheless such identification by FISH based techniques is the necessary identification of the structures as microbial cells, and the only one currently available.

To ask authors to complete such identification on the whole data set analyzed is not realistic as it is a huge amount of time and effort that may not even be possible to realize in decent amount of time.

However what can be done is to apply such identification followed by correlation with nanoSIMS imaging only on a couple of samples (small data set) and show that the cell-like particles positively identified by FISH and fluorescence microscopy imaging correspond in size and association with those previously identified and labelled by co-authors as N₂ fixing non-diazotrophs on nanoSIMS imaging only. This will be their missing prove. Otherwise the authors should refrain from using "cells" in the title and text and rather refer to those as "putative cells" or "cell-like particles".

2. Do you feel any requests are unfeasible or beyond the scope of the study?

I highly appreciate the effort that the authors invested in analyzing and processing such huge amount of data, particularly with the low throughput expensive instrument such as nanoSIMS. Also the effort that was put in re-processing the data with more conservative cut-offs in the revised version was a remarkable effort. The authors did a good job, also by answering most of the reviewers comments.

However the missing proven identification of these structures is a fact that can not be denied and is within the scope of the paper since the authors claim they have obtained "cell-specific measurements". The truth is that the authors just based on the ¹²C¹⁴N secondary ions (which is present in any biomass) positively identify ¹⁵N hot spots over a certain threshold in a mixed abundant biomass as "cells".

Regarding the requests of reviewer 2, I do not believe that revising the data again with a lower cutoff of 15% and using in addition to ¹²C¹⁴N also ³¹P or ³²S to "positively" identify cells will help much.

To shortly answer to the above question, no I do not believe the requests to provide prove for cell identification are unfeasible or beyond the scope of the study.

3. Is there anything you could recommend to the authors to help them address the concerns of reviewer #2?

Yes, as I mentioned above, at answer to question 1 (marked text above), FISH using group specific probe for non-diazotrophic N₂ fixers or even the general EUB probe will do.

Alternative is to use for title and text putative cells or cell-like particles

Reviewer #3 (Remarks to the Author):

I thank the authors for their new analyses and efforts in answering the comments and questions raised by the reviewers.

I am still not convinced that the approached used confirms, unambiguously, N₂ fixation in NCDs.

Some final comments below:

L168: ^{15}N or ^{13}C enrichment does not confirm direct uptake, it can be cross-feeding. Your additional analyses may confirm they are cells, but you cannot confirm they are diazotrophic.

L341: The authors now add:

"While it is possible the new N biomass synthesized by NCDs may be due to secondary uptake of ^{15}N labeled NH_4 derived from cyanobacteria, it is unlikely those values would be high enough to pass the conservative minimum ^{15}N threshold used to define an N_2 -fixing cell given the initial amount of ^{15}N added to the incubation ($2.8 \text{ At}\% \pm 0.4$)."

What is this threshold based on? The ^{15}N enrichment values shown in Fig. 2 are $\sim 0.7\%$, $\sim 1.5\text{-}2\%$ in Fig. 3 and up to $\sim 3\%$ in Fig. S3. These enrichment values are lower than fixed N_2 transfer enrichment values (up to $\sim 6\%$) previously measured in diatoms, *Synechococcus* and heterotrophic bacteria (Bonnet et al. 2016; Berthelot et al. 2016). Unfortunately, I doubt there is a threshold that can be used to rule out secondary uptake of label.

Response to previous Comment 4:

"Comment 4; Line 166: The natural ^{15}N enrichment of cells varies largely and non- $^{15}\text{N}_2$ incubated samples should have been used as a reference instead of *Pseudomonas*.

Response: We appreciate the reviewer's concern, but we believe use of the *Pseudomonas* reference cells more than accounts for difference in natural abundance of ^{15}N ."

I still do not agree. Natural ^{15}N abundance can change drastically over an incubation period of 24h, which is known to affect the detectability of low N_2 fixation rates (see work from Julie Granger and Ally Fong). However, the 0.47% threshold is conservative and likely within the natural variability that real NCDs experienced in this study. However, I strongly recommend measuring non- ^{15}N enriched incubated cells in future studies.

References

- Berthelot, Hugo, Sophie Bonnet, Olivier Grosso, Véronique Cornet, and Aude Barani. 2016. "Transfer of Diazotroph-Derived Nitrogen towards Non-Diazotrophic Planktonic Communities: A Comparative Study between *Trichodesmium erythraeum* *Crocospaera watsonii* and *Cyanothece* sp." *Biogeosciences* 13: 4005–21.
- Bonnet, Sophie, Hugo Berthelot, Kendra Turk-Kubo, Véronique Cornet-Barthaux, Sarah Fawcett, Ilana Berman-Frank, Aude Barani, et al. 2016. "Diazotroph Derived Nitrogen Supports Diatom Growth in the South West Pacific: A Quantitative Study Using NanoSIMS." *Limnology and Oceanography*. <https://doi.org/10.1002/lno.10300>.

Thank you for giving us the time and opportunity to resubmit a revised draft of our manuscript. We appreciate the time and effort the reviewers have spent in providing multiple rounds of feedback on our manuscript, (NCOMMS-21-19855) now titled “Direct cell-specific measurements show putative N₂ fixation by particle-attached non-cyanobacterial diazotrophs in the North Pacific Subtropical Gyre” for *Nature Communications*. We have incorporated all the reviewers’ suggestions and it has greatly improved our manuscript. The main change in the new version of the manuscript is the use of “putative cell” or “region of interest (ROI)” instead of the previously used “cell”. We have highlighted the changes within the manuscript with additional text highlighted in blue. Below is a point-by-point response to the reviewers’ comments and concerns, any line numbers included in the responses refer to the new version of the manuscript.

Thank you again for your consideration of our revised manuscripts.

REVIEWER COMMENTS

Reviewer #1 (Remarks to the Author):

I consider the authors have dedicated huge effort to improve and revise the manuscript. From my point of view they have addressed thoroughly all the raised concerns and criticism. I support the publication of the current revised version.

Response: We appreciate the time and effort you provided in reviewing our manuscript and thank the reviewer for recommending our study for publication.

Reviewer #2 (Remarks to the Author):

After the first round of review, I was quite hopeful that the authors would be able to provide the missing images and, with that, the missing evidence that the ‘cellular’ measurements are indeed from cells. Unfortunately, I must say that the revised version of the manuscript still falls short in providing the necessary evidence due to a lack of correlative imaging. The correlative imaging approach is the vital link between a cell and its activity, especially when cells are embedded in a matrix like, for example, the particles analyzed here. I do not think that the authors approach to use ¹²C/¹⁴N- ion counts to identify cells (even with the added constraints on the minor isotopes) is a well-founded approach. Even a combination of C,N,S and/or P imaging (as pointed out by reviewer 1) could have helped a little bit here, although some correlative imaging would have still be needed. As such, the only piece of evidence that argues for non-cyanobacterial, or any N₂ fixation on the particles for that matter, is that high enrichments of ¹⁵N can only come from N₂ fixation activity in the incubated water. But is this enough for publication?

Or should the authors maybe invest the time for some good-quality imaging that provides the evidence that measurements are indeed representative for cells, both for the NCD as well as all other cellular measurements (how can one see that the other ~41000 measurements are all cells and not just random biomass?). I think it would be great if it could be shown that particles are indeed sites of N₂ fixation but I

also think that the evidence needs to be a bit more convincing at this point (also considering some of the concerns mentioned by reviewers 1 and 3).

Response: Thank you for this explanation and we agree that our study does not provide unequivocal and definitive evidence that the enriched areas of ^{15}N are indeed cells, with this in mind we have edited all wording within the manuscript to reflect that uncertainty. The areas referred to as cells have been changed to putative cells or region of interest (ROI). Additionally, we have added a similar statement to the manuscript (Line 173) "Out of the 1000s of ROIs analyzed it is likely some percent are not cells but other organic matter from the complex organic particles they are associated with, therefore these will be referred to as ROIs rather than cells." Although we cannot definitively say the ^{15}N enriched areas are cells, the probability that they are cells remains high as only N_2 fixing organism could convert $^{15}\text{N}_2$ gas into ^{15}N enriched biomass.

Although we agree that correlative imaging would be a great addition to this study, currently there is no realistic means to achieve this goal for all NCD ROIs found. In the previous submission we were able to find and add figures of a particle with an associated NCD ROI using SEM which shows the N_2 enriched region has a clear outline with a morphology similar to what would be expected of a bacterial cell.

Alternatively, due to the destructive nature of nanoSIMS, finding a rare potential NCD ROI after nanoSIMS analysis is improbable due to the changing visual cues, and not possible when the ROI of interest is fully destroyed. Visual imaging before nanoSIMS is an unrealistic use of time as N_2 fixing ROIs were only found on 9 out of 1105 particle (1 out of 123) with many stations where none were found. Additionally, surface imaging of particles before nanoSIMS may not represent what is analyzed by nanoSIMS as a strong current presputter method is used to implant Cs^+ ions in the analysis area before data acquisition removing much of the surface layer within the frame of interest.

Visualization using specific 16S rRNA targeted FISH type probes is not currently feasible for NCDs as a group, as most NCDs are primarily known by the *nifH* gene. In theory the *nifH* gene or transcripts could be targeted by geneFISH or mRNA-FISH, which can be applied when only functional gene sequences are known, however, they are generally less efficient, more specific, and not well proven in this environment, thus, there is currently no good approach to target all NCDs present. As we and many others have demonstrated, marine NCDs are quite diverse excluding use of the few known NCD 16S rRNA genes to design probes to target NCDs as a whole group. The untargeted PCR-free based approach of our method ensures any ROIs fixing N_2 can be identified regardless of phylogenetic similarity to predetermined probes. We believe our study of putative NCD cells fixing N_2 is a critical starting point for future studies to take a more targeted approach linking cell identity and NCD N_2 fixation on particles.

Further comments:

I 171: Using a 30% error cut-off is fairly lenient/generous rather than conservative. Would the data and conclusion hold up if the authors used a more conservative cut-off of 5-10% (or maybe even 15%), which is mostly used in nanoSIMS studies? The authors could at least state how many 'cells' actually had errors that high, how many had errors quite far below the 30%?

Response: There is one NCD ROI (20% error) and one cyanobacterial-like ROI (23%) with an associated error value between 15 to 30% of the permil value. The average of the percent error relative to the permil value (associated error/permil value*100) is 8.1 +- 4.5% error for putative NCD cells, which is

similar to the cyanobacterial-like ROIs with an average percent error of 6.6 +- 4.3 overall. As one ROI per group would be lost (NCD and cyanobacterial-like), our conclusions are still valid as the majority of data would pass the lower percent error threshold (<15%) and the percent error of the NCD ROIs resembles the cyanobacterial-like ROI data which are known to fix N₂.

I 192: The authors determined the C and N content of relatively small cells using the equations provided by Verity et al. 1992. While these conversion factors are good for larger cells, small cells tend to have a higher C density (Khachikyan et al. 2019). Using the higher C content (and with that, also N content), more reflective of the bacterial biomass analyzed here, the cellular rates would be somewhat higher.

Response: Thank you for the comment, we have changed the N content equation used for small ROIs to the reviewer's suggestion based on Khachikyan et al., 2019. The putative cell-specific N₂ fixation rate plot in Fig. 3 and all subsequent calculations and discussion have been edited accordingly. The new equation has been noted in the methods section (Line 199) "N content for ROIs with spherical cell volume $\leq 0.6 \text{ } \mu\text{m}^3$ were estimated according to Khachikyan et al. (2019) as small can have proportionally higher N content than larger cells." The use of the more accurate N content per cell equation (Khachikyan et al., 2019) for our small ROIs only slightly altered the putative NCD cell-specific N₂ fixation rate by an average of $0.03 \text{ fmol N cell}^{-1} \text{ day}^{-1}$. Additionally, when using the nanoSIMS single cell N₂ fixation rate values of NCD ROIs to scale to potential volumetric N₂ fixation by NCDs the Khachikyan et al., 2019 equation resulted in little change in overall value from 0.011 to $0.0099 \text{ nmol N cell}^{-1} \text{ day}^{-1}$.

Figures 2, 3 and Supp. Fig 3: A threshold for the exclusion of data that goes beyond 'normal' processing seems to have been applied; however, this is not mentioned in the methods and/or figure legends. Regular nanoSIMS imaging does not produce the smooth edges seen here. Please add a disclaimer and/or, preferably, show the less-processed data.

Response: We thank the reviewer for bringing this to our attention and agree that the less processed data would be a better representation of NanoSIMS images. We have updated figures 2, 3 and Supp. Fig 3 accordingly. Particles are now outlined with red to show the particle shape and size as well as including secondary electron images of the particles for reference.

Reviewer #1 was asked to comment on the report from reviewer #2. These are their remarks:

1. Do you disagree with any of their technical criticisms?

I believe reviewer 2 criticism is fully justified. Presently the authors did not provided direct evidence that the small isotopically enriched spots/particles are actually cells. Even with the reviewer 2 recommendations to use a more conservative cut-off (15%) this situation will not change. It will probably just decrease the number of enriched cell-like particles considered in the analysis. The direct link of function to cell identity can be only solved by combination of FISH with nanoSIMS. However it is almost impossible to apply such correlation to the huge number of samples the authors analyzed (over

41,000 measurements).

The explanation of the authors against FISH identification that may result in a dilution of the isotopic enrichment of the cell-like particles is just. It was shown also previously, that FISH based identification decrease the isotopic enrichment and results in underestimation of the real cellular activity/uptake rate. Nevertheless such identification by FISH based techniques is the necessary identification of the structures as microbial cells, and the only one currently available.

Response: As noted in response to Reviewer 2, FISH approaches are not possible at this time for all NCD cells, which are uncultivated and known only by *nifH* gene sequences, not the 16S gene that would allow CARD-FISH to hybridize with rRNA. As a result, it is not possible to visualize all NCDs by FISH, which is the reason we took the approach in this study. In theory the *nifH* gene could be targeted by the more difficult, less efficient geneFISH or similar method which targets individual genes but the divergent identities present with the *nifH* genes of NCD prevent any designed probes from targeting all NCDs present. Additionally, the group is remarkably diverse excluding use of the few known 16S rRNA genes to design probes to target NCDs as a whole group. We believe the untargeted approach of our method is one of the strengths of the study as it ensures any cell that is fixing N₂ can be identified regardless of phylogenetic similarity to predetermined probes, or isotopic dilution caused by lengthy preprocessing of samples such as with FISH methods. We believe our study of putative NCD cells fixing N₂ will be a great starting point for future studies to take a more targeted approach linking cell identity and NCD N₂ fixation on particles. This approach is so far the only one that has the potential to answer the question: Do NCDs contribute significantly to oceanic N₂ fixation.

To ask authors to complete such identification on the whole data set analyzed is not realistic as it is a huge amount of time and effort that may not even be possible to realize in decent amount of time. However what can be done is to apply such identification followed by correlation with nanoSIMS imaging only on a couple of samples (small data set) and show that the cell-like particles positively identified by FISH and fluorescence microscopy imaging correspond in size and association with those previously identified and labelled by co-authors as N₂ fixing non-diazotrophs on nanoSIMS imaging only. This will be their missing prove. Otherwise the authors should refrain from using "cells" in the title and text and rather refer to those as "putative cells" or "cell-like particles".

Response: Thank you for the suggestion, we agree and have altered all mentions of "cell" to read putative cells or ROIs where appropriate. We provided SEM images and images of DAPI stained cells associated with particles and are convinced this evidence shows cells are associated with particles in our study. The use of general bacterial FISH probes, we believe, would not provide any more convincing information beyond what the DAPI stain cells already provided. We concur that identifying these as "putative cells" is a more accurate description for the data presented in this study and thank the reviewers for pointing that out.

2. Do you feel any requests are unfeasible or beyond the scope of the study?

I highly appreciate the effort that the authors invested in analyzing and processing such huge amount of data, particularly with the low throughput expensive instrument such as nanoSIMS. Also the effort that was put in re-processing the data with more conservative cut-offs in the revised version was a

remarkable effort. The authors did a good job, also by answering most of the reviewers comments.

However the missing proven identification of these structures is a fact that can not be denied and is within the scope of the paper since the authors claim they have obtained "cell-specific measurements". The truth is that the authors just based on the $^{12}\text{C}^{14}\text{N}$ secondary ions (which is present in any biomass) positively identify ^{15}N hot spots over a certain threshold in a mixed abundant biomass as "cells". Regarding the requests of reviewer 2, I do not believe that revising the data again with a lower cutoff of 15% and using in addition to $^{12}\text{C}^{14}\text{N}$ also ^{31}P or ^{32}S to "positively" identify cells will help much. To shortly answer to the above question, no I do not believe the requests to provide prove for cell identification are unfeasible or beyond the scope of the study.

Response: We thank the reviewer for understanding the difficulty in obtaining definitive evidence that our data are from cells. Although we agree that cell identification would be an informative addition to this study, for the reasons outlined above it is not possible for us to provide correlative imaging and nanoSIMS at this time aside from the already included DAPI stained images and SEM images. Due to this we agree "putative cell" is a more accurate description of the data presented in the manuscript.

3. Is there anything you could recommend to the authors to help them address the concerns of reviewer #2?

Yes, as I mentioned above, at answer to question 1 (marked text above), FISH using group specific probe for non-diazotrophic N_2 fixers or even the general EUB probe will do.

Response: Thank you for the suggestion but we do not think a general Bacterial FISH probe would show information beyond what we have already provided with images of the DAPI stained cells. A FISH approach would require either 1) generic probes for bacteria which are available, but would not discriminate between N_2 -fixing and non- N_2 -fixing taxa, or 2) specific probes for low abundance N_2 -fixers which would not necessarily catch the taxa that are fixing on particles anyway.

The first is likely to generate a number but it would be unclear how to correlate all bacteria in a particle sample to the N_2 -fixers on particles which are anticipated to be only a fraction of the bacteria on a subset of the particles. So the only information that would result is a confirmation that some bacteria can be seen on particles.

As we describe above in response to Reviewer #1's concerns, we note again that this second approach is not possible because marine NCDs are primarily known by *nifH* gene sequence, which cannot be used to develop CARD-FISH probes which target rRNA. Developing techniques to target specific groups of NCDs using geneFISH or mRNA-FISH cannot be reasonably accomplished in support of this study.

Reviewer #3 (Remarks to the Author):

I thank the authors for their new analyses and efforts in answering the comments and questions raised

by the reviewers.

I am still not convinced that the approach used confirms, unambiguously, N₂ fixation in NCDs. Some final comments below:

L168: ¹⁵N or ¹³C enrichment does not confirm direct uptake, it can be cross-feeding. Your additional analyses may confirm they are cells, but you cannot confirm they are diazotrophic.

Response: Thank you for the comment and we acknowledge that the question of measuring cross-feeding versus fixation is an issue in all mixed environmental samples incubated with isotopes and analyzed with nanoSIMS thus is not specific to our study. We intentionally chose short (24hr) incubations and set minimum thresholds of 0.47 At% ¹⁵N to mitigate this problem. Furthermore, the high values of ¹⁵N enrichment measured strongly indicates primary incorporation of the isotope by N₂ fixation. Measurable cross feeding would likely present as a higher number of putative cells with lower ¹⁵N enrichment rather than a few select ROIs with higher ¹⁵N enrichment.

L341: The authors now add:

“While it is possible the new N biomass synthesized by NCDs may be due to secondary uptake of ¹⁵N labeled NH₄ derived from cyanobacteria, it is unlikely those values would be high enough to pass the conservative minimum ¹⁵N threshold used to define an N₂-fixing cell given the initial amount of ¹⁵N added to the incubation (2.8 At% ± 0.4).”

What is this threshold based on? The ¹⁵N enrichment values shown in Fig. 2 are ~0.7%, ~1.5-2% in Fig. 3 and up to ~3% in Fig. S3. These enrichment values are lower than fixed N₂ transfer enrichment values (up to ~6%) previously measured in diatoms, *Synechococcus* and heterotrophic bacteria (Bonnet et al. 2016; Berthelot et al. 2016). Unfortunately, I doubt there is a threshold that can be used to rule out secondary uptake of label.

Response: We agree that it is impossible to set a threshold to completely rule out secondary uptake but we believe the conservative threshold chosen minimizes that possibility as far as is reasonable given the data presented.

The ¹⁵N At% minimum threshold for enriched ROIs is based on measurements of unenriched *pseudomonas* cells used as standard controls for the nanoSIMS instrument. The threshold is defined as the average At% plus 3 times the standard deviation of many standard cells from multiple measurements. This threshold can more than account for the natural variation of ¹⁵N among ROIs as well as the technical variation of the instrument runs and the inherent counting error associated with nanoSIMS measurements.

We thank the reviewer for mentioning several studies investigating secondary uptake of diazotroph-derived nitrogen (DDN) and upon further consideration of our text realized addition of more context of how our minimum threshold compares to such studies will be informative to a reader. Therefore, we have added the following text to the manuscript (line 354) “For context the minimum ¹⁵N threshold is at least 1.25x higher than the average uptake by plankton of diazotroph derived N during a cyanobacterial bloom (Bonnet et al., 2016; Bonnet et al., 2016b; Berthelot et al. 2016)”

We include a more detailed comparison here for the reviewer but would like to point out the studies highlighted were conducted during cyanobacterial diazotroph bloom conditions which would likely increase the amount of dissolved ^{15}N in the system compared to non-bloom scenarios such as were present at the time of our study. Regardless, our minimum threshold is approximately twice as high (0.47 At%) as maximum values reported for secondary uptake of diazotroph derived nitrogen (DDN) in Bonnet et al., 2016 after 24 hours (0.23 At%) with similar values of initial ^{15}N At% (2.3 At% this study and 2.4 At% Bonnet) in both studies. The lower initial At% in our study (2.3 At% highest value where NCDs were found) would likely result in slightly lower overall enrichment increasing the likelihood that our minimum threshold is higher than the large majority secondary uptake measurements found Bonnet et al., 2016. Comparison to two additional DDN secondary uptake studies also indicate our threshold was conservatively high to exclude the occurrence of secondary uptake being categorized as NCD N_2 fixation. In Bonnet et al., 2016B (VAHINE) DDN uptake was measured during a UCYN-C bloom with an average secondary enrichment value of 0.389 ± 0.014 after 24 hours for picoplankton with similar initial ^{15}N At% as ours (2.4 At% Bonnet and 2.3 At% this study). In the study by Berthelot et al. 2016 the authors simulated bloom conditions using cultures and found secondary uptake varies by diazotroph type with *Trichodesmium* allowing for the highest values of secondary uptake at 0.244 At% after 24 hours (48hr measurement/2) in non-pigmented bacteria. Although the initial ^{15}N At% present in this study was much higher (3.5 ± 0.2 At%) than ours (2.3%) and therefore less easily comparable, we would expect higher enrichment with higher initial ^{15}N At% which is further indication our threshold is reasonably high to rule out high proportions of miscategorized data.

Response to previous Comment 4:

“Comment 4; Line 166: The natural ^{15}N enrichment of cells varies largely and non- $^{15}\text{N}_2$ incubated samples should have been used as a reference instead of *Pseudomonas*.

Response: We appreciate the reviewer’s concern, but we believe use of the *Pseudomonas* reference cells more than accounts for difference in natural abundance of ^{15}N .”

I still do not agree. Natural ^{15}N abundance can change drastically over an incubation period of 24h, which is known to affect the detectability of low N_2 fixation rates (see work from Julie Granger and Ally Fong). However, the 0.47% threshold is conservative and likely within the natural variability that real NCDs experienced in this study. However, I strongly recommend measuring non- ^{15}N enriched incubated cells in future studies.

Response: Thank you for the advice, we agree with the reviewer and will make every effort in future studies to amend our experimental design to meet this standard.

References

- Berthelot, H., Bonnet, S., Grosso, O., Cornet, V., & Barani, A. (2016). Transfer of diazotroph-derived nitrogen towards non-diazotrophic planktonic communities: A comparative study between *Trichodesmium erythraeum* *Crocospaera watsonii* and *Cyanothece* sp. *Biogeosciences*, *13*(13), 4005–4021. <https://doi.org/10.5194/bg-13-4005-2016>
- Bonnet, S., Berthelot, H., Turk-Kubo, K., Cornet-Barthaux, V., Fawcett, S., Berman-Frank, I., ... Capone, D. G. (2016). Diazotroph derived nitrogen supports diatom growth in the South West Pacific: A quantitative study using nanoSIMS. *Limnology and Oceanography*, *61*(5), 1549–1562. <https://doi.org/10.1002/LNO.10300>

- Bonnet, S., Berthelot, H., Turk-Kubo, K., Fawcett, S., Rahav, E., L'Helguen, S., & Berman-Frank, I. (2016B). Dynamics of N₂ fixation and fate of diazotroph-derived nitrogen in a low-nutrient, low-chlorophyll ecosystem: Results from the VAHINE mesocosm experiment (New Caledonia). *Biogeosciences*, 13(9), 2653–2673. <https://doi.org/10.5194/bg-13-2653-2016>
- Khachikyan, A., Milucka, J., Littmann, S., Ahmerkamp, S., Meador, T., Könneke, M., ... Kuypers, M. M. M. (2019). Direct Cell Mass Measurements Expand the Role of Small Microorganisms in Nature. Retrieved from <https://doi.org/10>

REVIEWERS' COMMENTS

Reviewer #1 (Remarks to the Author):

The authors have been investing quite an effort in revising the current version of the manuscript addressing all points of criticism.

Although the proof of ROIs as microbial cells is missing, the explanation of the authors regarding the use of FISH for cell identification is understandable and justified. Changing the wording from "cells" to "putative cells" is in line with methodological limitations. In my opinion ^{15}N enrichment in such short incubation time is a strong indication of diazotrophy and likelihood of secondary metabolite transfer is not high.

I have no further comments. I support the publication of the revised version.

Reviewer #2 (Remarks to the Author):

The majority of my comments have been resolved/answered by the authors.

One aspect I am still really struggling with is the fact that the ^{15}N enrichment shown in the nanoSIMS cannot be linked to cells; it does not have to be linked to a certain identity of a cell, but just that it is a cell, for example, via DAPI/SYBRgreen (or similar staining) imaging or EUB-FISH imaging (and although the latter would be much more laborious than a nucleic acid stain, it would indeed show whether these are intact cells).

It is very unfortunate that the authors did not invest into an imaging technique prior to nanoSIMS, which is usually the prerequisite for linking any measured activity to a cell. I do not agree with the argument that it would take too much time; if one invests such expensive nanoSIMS measurement time, then there certainly should be time for normal fluorescence microscopy, in my opinion. If the simple fluorescence imaging had not worked out because of cells overlying each other in particles or a matrix, the next possible step would have been to embed particles in a resin and obtain thin or semi-thin sections that could have been properly imaged. Of course, this is all very laborious and, in fact, exemplifies the difficulties in unambiguously showing that a cell did indeed fix ^{15}N -labelled N_2 gas.

Now that the horse has left the barn, it obviously becomes very difficult to make the link between activity and cell. I sincerely hope that this oversight is prevented in the future.

Reviewer #3 (Remarks to the Author):

I am mostly satisfied with the authors' responses to my comments and the effort put in improving their text to be fairer with their results. In my opinion, what needs to be changed is the title, which promises to measure cell-specific NCD N_2 fixation, and it is not rigorously true. I acknowledge the immense challenge of developing an NCD-specific N_2 fixation measurement method, and the authors present here an important step ahead, which I'm sure they will complete in their upcoming publications. But just not yet. This is somewhat resolved by the addition of "putative", but I would recommend reconsidering "direct".

Thank you for giving us the time and opportunity to resubmit a revised draft of our manuscript. We appreciate the time and effort the reviewers have spent in providing multiple rounds of feedback on our manuscript, (NCOMMS-21-19855) now titled "Cell-specific measurements show putative N₂ fixation by particle-attached non-cyanobacterial diazotrophs in the North Pacific Subtropical Gyre" for Nature Communications. The main change in the new version of the manuscript is the addition of a discussion paragraph explaining possible alternative scenarios that could create cell-size ¹⁵N enrichments and possible methods options for future studies to link cells to the ¹⁵N enrichments on particles. We have highlighted the changes within the manuscript with additional text in blue. Below is response to the reviewers' comments and concerns, any line numbers included in the responses refer to the new version of the manuscript.

Thank you again for your time and comments.

REVIEWERS' COMMENTS

Reviewer #1 (Remarks to the Author):

The authors have been investing quite an effort in revising the current version of the manuscript addressing all points of criticism.

Although the proof of ROIs as microbial cells is missing, the explanation of the authors regarding the use of FISH for cell identification is understandable and justified. Changing the wording from "cells" to "putative cells" is in line with methodological limitations. In my opinion ¹⁵N enrichment in such short incubation time is a strong indication of diazotrophy and likelihood of secondary metabolite transfer is not high.

I have no further comments. I support the publication of the revised version.

Response: We thank reviewer 1 for their support, in addition to the time and helpful comments that improved our manuscript.

Reviewer #2 (Remarks to the Author):

The majority of my comments have been resolved/answered by the authors.

One aspect I am still really struggling with is the fact that the ¹⁵N enrichment shown in the nanoSIMS cannot be linked to cells; it does not have to be linked to a certain identity of a cell, but just that it is a cell, for example, via DAPI/SYBRgreen (or similar staining) imaging or EUB-FISH imaging (and although the latter would be much more laborious than a nucleic acid stain, it would indeed show whether these are intact cells).

It is very unfortunate that the authors did not invest into an imaging technique prior to nanoSIMS, which is usually the prerequisite for linking any measured activity to a cell. I do not agree with the argument that it would take too much time; if one invests such expensive nanoSIMS measurement time, then

there certainly should be time for normal fluorescence microscopy, in my opinion. If the simple fluorescence imaging had not worked out because of cells overlying each other in particles or a matrix, the next possible step would have been to embed particles in a resin and obtain thin or semi-thin sections that could have been properly imaged. Of course, this is all very laborious and, in fact, exemplifies the difficulties in unambiguously showing that a cell did indeed fix ^{15}N -labelled N_2 gas.

Now that the horse has left the barn, it obviously becomes very difficult to make the link between activity and cell. I sincerely hope that this oversight is prevented in the future.

Response: In order to address reviewer 3 concerns, we have added explanation to the discussion section explaining why our experiment was designed the way it was and the limitations that presents (Line 289) which the reviewer has highlighted. Although the untargeted approach can be a benefit as it does not cause isotope dilution through cell identification, we appreciate the reviewers concerns and note throughout the manuscript the possibility that ROIs have the potential to be organic material other than cells. Even though, we cannot link the ^{15}N enrichment directly to cells we believe our study will enable future studies to focus on particles which will minimize the amount of data needed to find these rare events and allow researchers to focus efforts on cell identification. As far as cell identification goes, we have added potential methods that could allow future studies to identify NCD on particles and link ^{15}N enrichment and cells on particles (Line 312).

Reviewer #3 (Remarks to the Author):

I am mostly satisfied with the authors' responses to my comments and the effort put in improving their text to be fairer with their results. In my opinion, what needs to be changed is the title, which promises to measure cell-specific NCD N_2 fixation, and it is not rigorously true. I acknowledge the immense challenge of developing an NCD-specific N_2 fixation measurement method, and the authors present here an important step ahead, which I'm sure they will complete in their upcoming publications. But just not yet. This is somewhat resolved by the addition of "putative", but I would recommend reconsidering "direct".

Response: We thank the reviewer for their comment and agree, the title no longer includes 'direct' and is now 'Cell-specific measurements show N_2 fixation by particle-attached putative non-cyanobacterial diazotrophs in the North Pacific Subtropical Gyre'.